# ANOMALY-GYM: A BENCHMARK FOR ANOMALY DETECTION IN EMBODIED AGENT ENVIRONMENTS

## ABSTRACT

Research on anomaly detection in reinforcement learning settings is sparse. Only a handful of methods have been proposed that - due to the absence of established evaluation scenarios - are evaluated on simple, small-scale, and self-proposed environments. This not only results in poor comparability but also leads to a limited understanding of the strengths and weaknesses of current approaches, rendering their applicability in real-world scenarios questionable. We address this problem by introducing Anomaly-Gym, a comprehensive evaluation suite for anomaly detection in reinforcement learning settings. In contrast to prior work, Anomaly-Gym is based on principled design criteria that disentangle evaluation from methodology. By enforcing specific constraints on the environments and anomalies considered, we propose a broad spectrum of evaluation data that covers both simulated and real-world tasks. In total, our benchmark features 10 different environments, 25 anomaly types, 4 strength levels, as well as multiple sensor modalities. We demonstrate the importance of these different aspects in a series of experiments on pre-generated datasets. For instance, we show that simple methods, while generally neglected in previous work, achieve near-perfect scores for settings with observational disturbances. In contrast, detecting perturbations of actions or environment dynamics requires more complex methods. Our findings also highlight current challenges with anomaly detection on image data and provide directions for future research.

## 1 INTRODUCTION

Anomaly detection (AD) is an essential component of safe and reliable machine learning (ML) systems (Hendrycks et al., 2021). It allows systems to initiate a conservative fallback policy or hand over to human control whenever anomalies are detected that can potentially lead to unsafe or erratic behavior (Nguyen et al., 2015; Amodei et al., 2016). Posing a long-standing problem in the field of ML, AD has been studied thoroughly in domains such as computer vision (Yang et al., 2024), robotics (Wellhausen et al., 2020), and healthcare (Šabić et al., 2021) applications.

However, the Reinforcement Learning (RL) domain has only witnessed a handful of methods that address AD. The field lacks publicly available benchmark datasets with challenging problems and well-defined evaluation criteria. As a result, the evaluation of existing work focuses on simpler, small-scale environments, often introduced in the same work as the corresponding methods. Poor comparability and a limited understanding of the current approaches' strengths and weaknesses are consequences, rendering their applicability questionable, especially in real-world scenarios.

In this work, we address this problem from the bottom up with the following contributions. First, we propose a general framework for evaluating AD within RL settings. Recognizing the potential bias in existing evaluation schemes, our framework encompasses a set of principled desiderata and is based on a clear connection to existing literature on AD. Second, we present Anomaly-Gym, a suite of 10 diverse tasks and 25 anomalies designed to rigorously test, evaluate, and compare different aspects of AD for RL (see Figure 1). In contrast to any existing work, Anomaly-Gym also incorporates meticulously tuned anomaly strength levels as well as real-world data. Anomaly-Gym focuses on embodied agent environments because they pose particularly important challenges in terms of safety. Third, we demonstrate the utility of Anomaly-Gym in a series of experiments in which we evaluate existing detection methods and baselines across the various environments, anomaly types, strength levels, and

observation modalities. Our results highlight the importance of differentiating between all these different aspects and reveal the limitations of current approaches on image observations. We also show that thresholding detectors on normal data is difficult and how this affects timely detection of anomalies.

Anomaly-Gym[1] along with all experiments and datasets[2] are made publicly available.

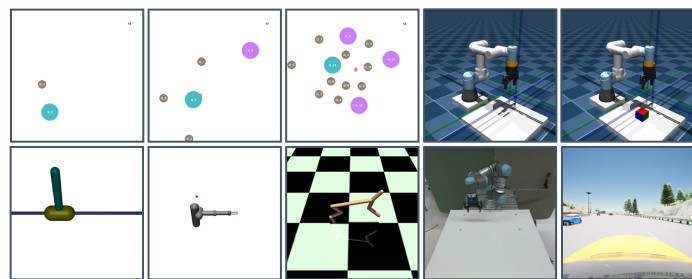

Figure 1: Overview of Anomaly-Gym environments.
*Top*: SAP-Goal{0, 1, 2}, URM-{Reach, Pick and Place (PnP)}
*Bottom*: MJC-{CartpoleSwingup, Reacher3D, HalfCheetah} URRtde-Reach, CAR-LaneKeep.

## 2 RELATED WORK

**Anomaly Detection in Reinforcement Learning.** In a relatively recent line of work, several techniques have been proposed that tackle AD specifically in RL settings. See Table 1 for an overview. While several benchmarks exist, they consider only small-scale settings, i.e., simple environments with small, discrete action spaces and low dimensional vector observations. As a result, state-of-the-art methods often disregard these benchmarks. Instead, new works typically introduce their own evaluation environments. This is problematic, as it can lead to distorted comparisons and potentially biased evaluation scenarios in favor of their own proposed methodology.

**Anomaly Detection in other Fields.** AD has been studied extensively in other fields. Although RL has unique characteristics that are important to consider, i.e., sequential-interactive data, many approaches from other fields can, in theory, also be adapted to RL. **AD for temporally independent** samples is a long-standing problem. See Chandola et al. (2009); Chalapathy & Chawla (2019) for surveys of classic and deep-learning-based methods. **AD for time-series** considers temporal components of the data. See Lai et al. (2021) for a detailed survey. **AD for image data** is also widely studied. Most commonly, classification tasks are considered, where the goal is to classify test samples within the label space and to reject samples with semantics outside its support (Yang et al., 2024). This task is also known as out-of-distribution (OOD) detection. **Video AD** considers temporal sequences of images. See Yang et al. (2024); Nayak et al. (2021) for recent surveys. **Robotics** and RL are also inherently related. However, methods from this domain are typically specialized for individual tasks or robotic platforms. Nonetheless, methods such as Hornung et al. (2014); Wellhausen et al. (2020); Ji et al. (2022) are potentially also applicable to AD in RL.

In summary, all of the above fields differ in their specification of problem setup, assumptions, and data, but they share important features with the problem of AD for RL. Although theoretically possible, translation of these findings and methods towards AD in RL has not yet been practically adopted. We believe this is partly due to the absence of a public and comprehensive benchmark. Our work aims to bridge this gap by offering a coherent evaluation framework that is reproducible, comparable, and easy to use, thereby facilitating the transfer of knowledge from related research areas to RL settings. Offering a wide spectrum of different tasks, anomalies, and sensor modalities, the Anomaly-Gym evaluation suite and datasets allow for an in-depth analysis of existing techniques on one hand, and lay the foundation for the development of novel approaches on the other.

**Related Benchmark suites.** Beyond anomaly detection, several benchmark suites target robustness and generalization in RL, aiming at the related goal of training robust policies that perform reliably under various disturbances. Procgen Cobbe et al. (2020) evaluates agents under procedural environment generation to test generalization across unseen levels. The Real-World RL suite Dulac-Arnold et al. (2020) emphasizes challenges such as delayed rewards, safety constraints, and stochastic dynamics to assess policy reliability under perturbed conditions. The DMControl generalization benchmark Hansen & Wang (2021) focuses on vision-based visual domain shifts (color and background augmentations). However, these suites lack the necessary infrastructure to evaluate

---

[1]Code: https://anonymous.4open.science/r/iclr-18811
[2]Datasets: https://www.kaggle.com/datasets/anonymous31459/anomaly-gym

Table 1: Related work on AD for RL. **Anomalies**: logic specific to gridworlds (L), action- (A), observation- (O), dynamics- (D) disturbances. **Observation Space**: Vector(V) or Image(I) based. **Action Space**: Discrete or Continuous. **Real:** real-world data. **Pub:** data/code publicly available.

| Ref. | Category | Environments | Anomalies | | | | Obs. | | Act. | | Real | Pub |
|---|---|---|---|---|---|---|---|---|---|---|---|---|
| | | | L | A | O | D | V | I | $\mathbb{N}$ | $\mathbb{R}$ | | |
| Sedlmeier et al. (2019) | Method | Classic Gym, Gridworlds | ✓ | | | | ✓ | | ✓ | | | |
| Sedlmeier et al. (2020) | Method | Classic Gym | | | | ✓ | ✓ | | ✓ | | | |
| Mohammed & Valdenegro (2021) | Benchmark | Classic Gym | | | | ✓ | ✓ | | ✓ | | | |
| Goel et al. (2021) | Benchmark | Gridworlds | ✓ | | | | ✓ | | ✓ | | | ✓ |
| Müller et al. (2022) | Conceptual | | | | | | | | | | | |
| Balloch et al. (2022) | Benchmark | Gridworlds | ✓ | | | | ✓ | | ✓ | | | |
| Danesh & Fern (2021) | Method | Classic Gym, Pybullet Ctrl. | | ✓ | | | ✓ | | | ✓ | | ✓ |
| Haider et al. (2023) | Method | Mujoco Ctrl. | | ✓ | | ✓ | ✓ | | | ✓ | | ✓ |
| Nasvytis et al. (2024) | Method | Classic Gym | | | | ✓ | ✓ | | | ✓ | | ✓ |
| Martinez et al. (2024) | Analysis | Gridworlds | ✓ | | | | ✓ | | ✓ | | | ✓ |
| Haider et al. (2024) | Analysis | PickAndPlace | | | | ✓ | ✓ | | | ✓ | | |
| Zollicoffer et al. (2024) | Method | Gridworlds | | | ✓ | ✓ | ✓ | | ✓ | | | |
| Zhang et al. (2024) | Method | Classic Gym, Atari, Carla | | | ✓ | | ✓ | ✓ | ✓ | | | |
| ours | Benchmark | Mujoco Ctrl., Carla,Particles, Robot manip. | | ✓ | ✓ | ✓ | ✓ | ✓ | | ✓ | ✓ | ✓ |

anomaly detection. Anomaly-Gym complements these efforts by providing the necessary infrastructure—specifically, controlled, labeled anomaly processes (sensor, actuator, and dynamics perturbations with calibrated strengths and timed onsets), reproducible data-generation pipelines, and standardized evaluation protocols.

# 3 ANOMALY DETECTION IN REINFORCEMENT LEARNING

To establish a clear and precise connection between AD and RL contexts, we start with a formal definition of the problem and review the key taxonomy adopted in this work.

## 3.1 ANOMALY DETECTION

**Definition of Anomaly.** *An anomaly is an observation that deviates considerably from some concept of normality* (Chandola et al., 2009).

Following Ruff et al. (2021), this can be formulated more formally via probability theory. Let $\mathcal{X} \subseteq \mathbb{R}^D$ represent the data space associated with a specific task and let $P$ be a probability distribution over $\mathcal{X}$. We define the notion of normality as a probability distribution $P^+$ over $\mathcal{X}$. An anomaly is then an observation $x \in \mathcal{X}$ that resides in a low-probability region under $P^+$ such that

$$\mathbf{A} = \{x \in X \mid p^+(x) \leq \epsilon\}, \tag{1}$$

where $p^+$ is a pdf of $P$ and $\epsilon \geq 0$ is some threshold.

**Types of Anomalies.** Several anomaly types have been defined in the literature (Chandola et al., 2009). A point anomaly refers to individual anomalous samples $x \in \mathbf{A}$. A group anomaly is a collection of related samples, where the group as a whole exhibits anomalous behavior. A contextual anomaly refers to samples that appear anomalous within a specific context, e.g., time or space.

**Terminology.** While anomaly, outlier, novelty or Out-of-Distribution (OOD) samples are often distinguished, they fundamentally refer to low-probability samples under $P^+$ (Ruff et al., 2021).

Consequently, methods for detecting such instances are inherently the same, regardless of the term (OOD, outlier, novelty, anomaly). Therefore, we use the umbrella term *anomaly*.

## 3.2 Connection to Reinforcement Learning

**Reinforcement Learning.** In RL, we consider sequential decision-making problems. Formally, this can be described as a discrete-time Markov Decision Process (MDP) (Puterman, 2014). An MDP is defined by the tuple $\mathcal{M} = (\mathcal{S}, \mathcal{A}, \mathcal{T}, r)$, where $\mathcal{S}$ denotes the state space, $\mathcal{A}$ the action space, $\mathcal{T} : \mathcal{S} \times \mathcal{A} \to \mathcal{S}$ the transition operator that describes the system dynamics, and $r : \mathcal{S} \times \mathcal{A} \to \mathbb{R}$ is the reward function. The RL objective is to find a policy $\pi_\theta : \mathcal{S} \to \mathcal{A}$, parameterized by $\theta$, which selects actions that maximize the expected cumulative sum of future rewards.

**Anomaly Detection in Reinforcement Learning.** The interaction between policy and MDP is the fundamental data-generating process in RL. Assuming the policy is fixed after training [3], anomalies in an MDP can be described via perturbations to individual components thereof, e.g., perturbations to the state space, the action space, or the transition dynamics (Haider et al., 2021). Let $\Gamma : \mathcal{M} \mapsto \mathcal{M}$ be a perturbation to one or more components of an MDP. The objective of AD in RL is to identify whether a given sample $x$ originates from the original MDP, $\mathcal{M}^+ \equiv \mathcal{M}$, or from an anomalous MDP, $\mathcal{M}^- \equiv \Gamma(\mathcal{M})$. Following the same argumentation as above, the problem of AD in RL reduces to

$$\mathbf{A} = \{x \in X \mid p_{\mathcal{M}^+}(x) \leq \epsilon\}. \tag{2}$$

**Data in Reinforcement Learning.** Data in RL is fundamentally contextual and collective. Following some policy $\pi_\theta : \mathcal{S} \to \mathcal{A}$ on the MDP $\mathcal{M}$ for $T \in \mathbb{N}$ steps, trajectories are generated:

$$\tau_{\pi,\mathcal{M}} = \{s_0, a_0, \ldots, s_T, a_T\}, \tag{3}$$

where $a_t \sim \pi(\cdot|s_t)$ and $s_{t+1} \sim \mathcal{T}(\cdot|s_t, a_t)$. Observations and actions within a trajectory are temporally correlated and dependent on the policy and the MDP. Hence, anomalies in RL must be understood within this context. They can occur as individual, contextual, or group events, requiring consideration of anomalies at different levels: single states $x = s$, individual transitions $x = (s, a, s')$, or entire trajectories $x = \tau$.

**Data Paradigm.** Training RL agents requires interaction data with the normal MDP. Hence, access to data generated under normal, non-anomalous conditions can be assumed. This does not hold true for anomalous data, as anomalies are, by definition, rare, unpredictable, and, most importantly, unknown during the training phase. This scenario corresponds to an unsupervised setting with a contamination rate $\eta = 0$, i.e., the train data is assumed to consist entirely of normal samples (Aggarwal, 2017). While settings with $\eta > 0$ are possible, we do not consider them in this work.

## 4 Evaluating Anomaly Detection in RL

Following the aspects above, we propose a general framework that embeds AD into RL settings.

### 4.1 Framework for Anomaly Detection in Reinforcement Learning

We motivate our evaluation framework by practical considerations. Reinforcement learning agents are typically trained with data that is representative of the deployment MDP under *normal* conditions. However, during deployment, agents can encounter substantially different inputs. To address this, we aim to monitor interactions during deployment and identify samples that deviate significantly from the learned model of normality, allowing for adjustments to the mode of operation, i.e., hand over to human operator or transition to safe state.[4] Hence, this framework consists of 4 stages:

**Stage 1) - Agent Training.** The RL agent interacts with the environment to learn a policy through trial and error. Data from early stages of training can differ drastically from later stages and is thus not stored for purposes other than training the agent. The policy will be fixed after this stage.

---

[3]We presume that the MDP itself - and consequently all data generated by interacting with it - can be subject to anomalies, but not the policy. While the policy could also change, we do not consider this case, as we believe it is less common in practice.

[4]This work focuses on the detection of anomalies; mitigation strategies are out of scope.

**Stage 2) - Data Generation.** Training data is generated by applying the policy to the normal MDP. This is equivalent to storing data during the final stages of agent training.

**Stage 3) - Detector Training.** Anomaly detectors are trained on normal data (model of normality).

**Stage 4) - Evaluation/Deployment.** The policy interacts with a potentially anomalous MDP. Anomalies can emerge at random time points (random onset). The goal of the detector is to identify anomalous samples as soon as they occur.

In settings that require online policy adaptation, the same data that is used to update the policy, can be used to update the detector online. This is equivalent to applying this framework iteratively:

1) Train agent → 2) collect data → 3) train policy & detector → 2) collect data → 3) (re)-train ...

*Connection to Sequential Decision Making:* More general, any policy/controller (e.g. offline RL policy, classic controller) can be plugged into this framework - the framework itself is policy agnostic. In this work we concentrate on RL. More details are provided in the discussion.

### 4.2 DESIDERATA TOWARDS EVALUATION DATA

Following our framework from Section 4.1, anomalous data emerge from perturbations to the normal MDP. To generate datasets for AD in RL, the normal environment and perturbations to this MDP (anomalous versions of the normal environment) are required. In the following, we define a series of essential desiderata towards both. Following these criteria should result in comprehensive datasets that can be used to analyze and compare different approaches for AD in RL scenarios.

**Environment Desiderata**

- **ED1-Diversity.** Environments should cover a wide range of scenarios and complexities to test the general applicability of AD methods. This includes diverse sensor and actuator modalities.
- **ED2-Scalability.** Environments should cover varying sizes to test the scalability of methods.
- **ED3-Realism.** Environments should incorporate realistic settings to ensure that detection methods are applicable in real-world scenarios. This includes continuous observations and actions.
- **ED4-Solvability.** Environments should allow RL systems to achieve (partial) success. This ensures that meaningful and non-trivial regions of the state space are reached.
- **ED5-Reproducibility.** Environments should be reproducible for consistent evaluation results.
- **ED6-Configurability.** Environments should allow customization of parameters.

**Anomaly Desiderata**

- **AD1-Diversity.** Anomalies should encompass different types to broadly evaluate detection capabilities across different failure modalities.
- **AD2-Realism.** Anomalies should mimic realistic faults or unexpected behaviors within the environment to ensure applicability in realistic settings.
- **AD3-Impact.** Anomalies should have varying levels of impact on the environment to evaluate detection capabilities across a spectrum of disruptions.
- **AD4-Difficulty.** Anomalies should be non-trivial to detect, exhibiting characteristics similar to normal operation. For instance, extreme sensor values or shutdown (e.g., a full black/white image) would be trivial to detect.

## 5 ANOMALY-GYM

We present Anomaly-Gym, a suite of sequential decision-making problems specifically designed to evaluate AD in RL. To satisfy the above-described desiderata, Anomaly-Gym includes and implements the following environments and anomalies.

### 5.1 ENVIRONMENTS

**MuJoCo Control (MJC)** The robotics control tasks from Brockman (2016) serve as widely adopted benchmark for RL algorithms. We include *Cartpole-Swingup*, *Reacher3D*, and *HalfCheetah*.

**Single Agent Particle Env (SAP)** is a set of three simple navigation environments, where the agent controls a particle to reach a goal while avoiding collisions with obstacles. The idea behind this is to mimic existing Grid-world scenarios but with more complex, vector-based (lidar) observations and continuous actions. We developed three levels of difficulty, called *Sape-Goal-{0,1,2}*.

**Universal Robots MuJoCo (URM)** is a set of two different robotic manipulation tasks based on a model of the Universal Robots UR3 in MuJoCo (Todorov et al., 2012). We implemented a simpler *Reach* task and a more complex *Pick-And-Place(PnP)* task.

**Universal Robots Rtde (URRtde)** is a robotic manipulation environment using a real-world Universal Robots UR3 and an RTDE interface (Lindvig et al., 2025). We implemented a simple *Reach* task with this environment, that mimics the mujoco simulation.

**Carla Lanekeep (CAR)** is an autonomous driving environment, where the agent controls a vehicle such that it stays in its lane, keeps a safe distance from other vehicles, and drives at a target speed on a highway. We implemented this environment using CARLA (Dosovitskiy et al., 2017).

All environments are described in more detail in Appendix A.1

## 5.2 ANOMALIES

**Observation Anomalies** are perturbations to the observations emitted by the environment. We implement the following observation anomalies:

Noise: $o'_t = o_t + e_t$     Scaling: $o'_t = o_t \cdot \beta$     Offset: $o'_t = o_t + \beta$

Drift: $a'_t = a_t + \beta * t$     Quantization: $o'_t = \beta \cdot \left\lfloor \frac{o_t}{\beta} \right\rfloor$     Temporal Noise: $o'_t = o_t + \beta\, n_{t-1} + \varepsilon_t$

where   $\varepsilon_t \sim \mathcal{N}(0, \beta^2)$,   $n_0 = \varepsilon_0$ and $\beta \in \mathbb{R}^+$.

**Action Anomalies** are perturbations to the actions before they are applied to the environment. We implemented the same types of anomalies as those used for observations but instead of quantization, we add action delay: $o'_t = o_{t-\beta}$ where $\beta \in \mathbb{N}^+$

**Dynamics Anomalies** refer to perturbations in the underlying dynamics function of an MDP, which makes them environment-specific. For example, we implemented moving objects for SAP, changed friction parameters in MJC, or applied disturbance forces in URM.

All anomalies are described in more detail in Appendix A.1 and Appendix A.2.

## 5.3 ANOMALY STRENGTHS

What constitutes a light anomaly and what a strong anomaly is not trivial to define. For instance, multiplying the mass of a robot joint by some factor can lead to an entirely different effect than multiplying the policy action by the same factor. Vice versa, the exact same anomaly can lead to vastly different results in two different environments. To enable a quantitative comparison of different anomalies in different environments we propose the following process.

Let $J_{\mathcal{M}^+}(\pi_\theta) = \mathbb{E}[\sum_{t=0}^T r_t]$ be the average cumulative reward of policy $\pi_\theta$ in the normal MDP, where $r_t$ is the reward received at time step $t$ and $T$ is the time horizon. Let $J_{\mathcal{M}^+}(\pi_R)$ be the average cumulative reward of a random policy, and $J_{\mathcal{M}^-}(\pi_\theta)$ the reward in the anomalous MDP. To this end, we define policy degradation through the normalized score

$$\bar{J}_{\mathcal{M}^-}(\pi_\theta) = \frac{J_{\mathcal{M}^-}(\pi_\theta) - J_{\mathcal{M}^+}(\pi_R)}{J_{\mathcal{M}^+}(\pi_\theta) - J_{\mathcal{M}^+}(\pi_R)}. \tag{4}$$

Using this normalized score, we can tune the magnitude of all anomalies and define different strength levels with respect to the degree of degradation of the rollout policy. We set four different levels of anomaly strength, namely *tiny* ($\bar{J}_{\mathcal{M}^-}(\pi_\theta) \approx 0.99$), *medium* (0.9), *strong* (0.75), and *extreme* (0.50).

To find the respective parameters, we conducted a grid search for each anomaly, strength and environment combination. Since this is an inherently noisy process, we collected 500 episodes for each grid point [5]. More details are available in Appendix A.4.

---

[5]For some anomalies not all strength levels can be reached. We omit those settings during evaluation.

## 5.4 SATISFACTION OF DESIDERATA

In the following we briefly describe how the above described environments and anomalies meet the defined desiderate. More details are available in Appendix A.3

- **ED1-Diversity.** In total we include 10 different environments from 4 different fields: Autonomous Driving(CAR), Robotics Manipulation (URM), Control (MJC) and navigation (SAP).
- **ED2-Scalability.** Observation sizes range from $\mathcal{O} \in \mathbb{R}^4$ to $\mathcal{O} \in \mathbb{R}^{26}$ for vector observations or $\mathbb{R}^{3x256x256}$ for image observations. Action spaces range from $\mathcal{A} \in \mathbb{R}^2$ to $\mathcal{A} \in \mathbb{R}^7$.
- **ED3-Realism.** URRtde is a real world Environment.
- **ED4-Solvability.** We train a policy until a success rate of ¿90 % is reached in each environment. Success is defined as follows per environment
- **ED5-Reproducibility.** Initial conditions are controlled by consistent random seeding.
- **ED6-Configurability.** All environment parameters can be adjusted.
- **AD1-Diversity:** We cover three fundamental anomaly types—observation (sensor faults), action (actuator faults), and dynamics (environmental changes)—with 6, 6, and 3 subtypes respectively.
- **AD2-Realism:** Observation anomalies mirror real sensor degradation (drift, noise, quantization errors). Action anomalies represent actuator failures (delays, scaling errors, temporally correlated noise). Dynamics anomalies model physical changes (friction degradation, mass shifts, external forces) encountered in real deployments.
- **AD3-Impact:** Our calibration process (Section 5.3) ensures anomalies span four impact levels (tiny to extreme) based on policy degradation, enabling systematic difficulty assessment.
- **AD4-Difficulty:** We explicitly exclude trivial cases (e.g., sensor shutdown, full image corruption). All anomalies maintain plausible system behavior—tiny anomalies cause <1

## 5.5 PRE-GENERATED DATASETS

We also provide a set of pre-generated data, which we collected by following the presented framework from Section 4.1. For this, we first trained a policy using TQC (Kuznetsov et al., 2020) until a success rate $> 95\%$ is achieved. This heuristic is environment-specific (e.g target speed without collision in *CAR*, place object on target in *URM-PnP*). To reach this success level, we performed a hyperparameter optimization, searching along the most important parameters (learning rate, network size, batch size). More details on agent training and hyperparameters are provided in Appendix A.6.

The final policy is used to generate train and test data for AD. Train data consists of $N$ normal episodes, containing only transitions generated with the normal MDP: $\mathcal{D}_{\text{train}} = \{\tau_{\pi,\mathcal{M}^+}\}_{n=1}^N$.
Test data consists of normal and anomalous episodes: $\mathcal{D}_{\text{test}} = \{\tau_{\pi,\mathcal{M}^+}\}_{n=1}^N \cup \{\tau_{\pi,\mathcal{M}^-}\}_{n=1}^N$.
Anomalous episodes are generated by introducing a perturbation after a randomized number of steps $t_a \in (t_0, t_H)$ (random onset). The timesteps $[t_0, ..., t_{\alpha-1}]$ are labeled as normal, whereas $[t_a, ..., t_H]$ are anomalous. The resulting dataset is balanced in expectation. For *Carla* and *URRtde*, we collect $N = 50$ episodes in each setting, for all other environments $N = 100$, with different random seeds.

## 6 EXPERIMENTS

To assess our benchmarks utility and the importance of its different aspects, we perform an empirical study with relevant baseline and *state-of-the art* anomaly detectors for RL.

## 6.1 DETECTION METHODS & BASELINES

We compare the following methods for vector observations: **IF** Isolation Forest (Liu et al., 2008), **KNN** k-Nearest Neighbor (Cover & Hart, 1967), **OCSVM** One-Class Support Vector Machine (Schölkopf et al., 2001), **RIQN** Recurrent implicit quantile network (Danesh & Fern, 2021), **MLP-DM** Dynamics model (DM) with an MLP as backbone, **LSTM-DM** DM with LTSM as backbone, and **PE-DM** Probabilistic ensemble DM (Haider et al., 2023).

For image observations we consider: **AE** Auto-Encoder reconstructing the current frame, **PredAE** Auto-Encoder predicting the next frame, **KNN-AE** KNN on latent features of an AE, **KNN-ResNet**, deep-KNN on latent features of a ResNet pre-trained on classification, similar to Sun et al. (2022),

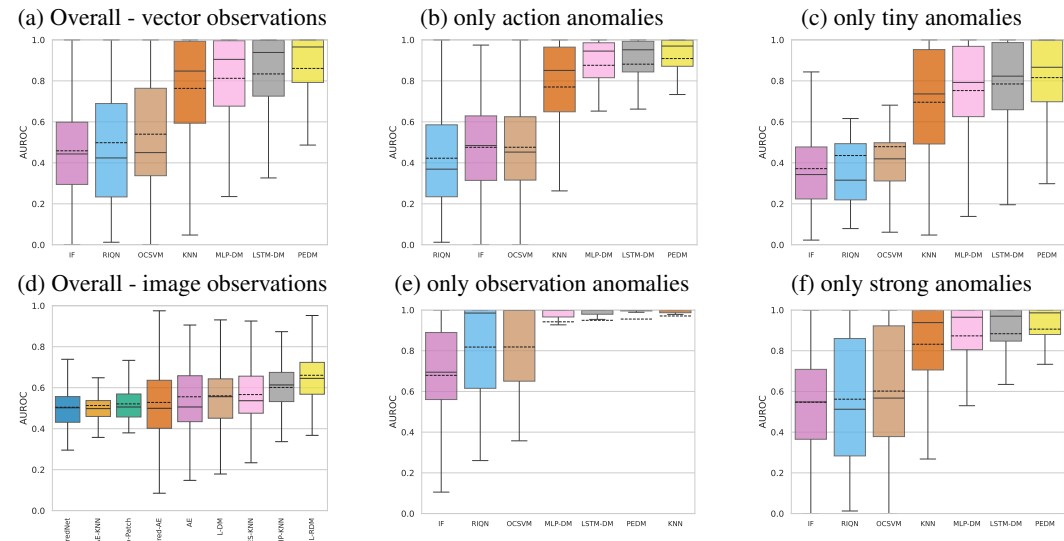

Figure 2: Distribution of AUROC ↑ scores for different detectors. Overall: all environments, anomalies and strengths. Other: all environments but only subgroup of anomaly type/strength. Detectors are ordered from left to right by average AUROC (dashed line).

**PredNet** (Lotter et al., 2017) predicting future frames, **Dino-PatchCore** Dino Patches with max. distance to NN, a simplified version of Roth et al. (2022), and **ClipKnn**: KNN search on CLIP embeddings, similar to Sun et al. (2022) but with CLIP embeddings (Radford et al., 2021). Furthermore, we introduce two additional baselines: **LDM** based on a latent DM similar to Haider et al. (2023), but with an AE and **LRDM**, with an additional Recurrent NN in latent space.

Note that we focus on external, policy agnostic methods. Techniques such as Sedlmeier et al. (2019) are thus not considered here. More details on all available methods can be found in Appendix A.7.

## 6.2 METRICS

We employ established metrics for anomaly detection:

- **AU-ROC** (Area Under Receiver Operating Characteristic curve): Measures overall discriminative ability across all thresholds
- **AU-PR** (Area Under Precision-Recall curve): Emphasizes performance on the positive (anomaly) class and thus suited for imbalanced datasets where anomalies are rare
- **FPR95:** False positive rate when detecting 95% of anomalies
- **VUS-ROC/VUS-PR:** address impacts of time-lags in auroc/aupr from (Paparrizos et al., 2022)

We study timing separately and hence we only report the AU-ROC scores in our main results. Detailed results including all metrics, are available in Appendix A.12. Note on calculation of Scores: there are two different ways to compute AUROC/AUPR/FPR95 in time-series:

- **Local**: computed per sequence (or per time series), then averaging over all sequnces
- **Global**: computed over the entire dataset by pooling all predictions and labels.

VUSPR/VUSROC on the other hand can only be calculated locally, as they consider time-series windows. We did not see any significant difference between local/global metrics and therefore reported only local metrics in the following. Global metrics are provided in the appendix Appendix A.12.

## 6.3 EVALUATION

**Overall detection performance.** To asses overall detection performance we consider the distribution of AUROC scores across all environments, anomaly types and strength levels. As reported in

Table 3: Influence of Rollout Policy - Comparison of SAC, TD3 and TQC

(a) Norm. Score: strong anomalies

| env | SAC | TD3 | TQC |
|---|---|---|---|
| MJC-Cartpole | 0.894 | 0.893 | 0.967 |
| SAP-Goal0 | 0.948 | 0.897 | 0.957 |
| URM-Reach | 0.840 | 0.810 | 0.940 |

(b) AUROC: Vec. Observations

| detector | SAC | TD3 | TQC |
|---|---|---|---|
| MLP-DM | 0.867 | 0.859 | 0.866 |
| LSTM-DM | 0.899 | 0.902 | 0.887 |
| PEDM | 0.918 | 0.926 | 0.910 |

(c) AUROC: Img. Observations

| detector | SAC | TD3 | TQC |
|---|---|---|---|
| AE | 0.634 | 0.655 | 0.613 |
| RES-KNN | 0.659 | 0.657 | 0.623 |
| CLIP-KNN | 0.663 | 0.663 | 0.635 |

Figure 2 a), methods that model environment dyanmics (MLP-, LSTM-, PE-DM) lead the ranking for vector observations. KNN, achieves the highest scores among all classic baselines and even outperforms one neural-network-based approach.

For image observations, overall detection performance is significantly lower and differences between detectors are marginal (Figure 2 d)). This shows that AD from images is a largely unsolved problem. LRDM achieves the highest scores and outperforms ViT-based approaches. LRDM constructs latent representations from the joint hidden state of observations and actions, thereby explicitly capturing environment dynamics. By contrast, the ViT baselines considered in this study operate on step-wise observations and consequently lack temporal dependencies. This highlights that incorporating RL-specific inductive biases (sequential dynamics) can substantially enhance detection performance.

**Analysis on Anomaly Types.** Figure 2 b) and e) show detection performance for action and observation anomalies respectively. For observation anomalies however, KNN achieves near perfect scores, slightly outperforming all other methods. For action anomalies, dynamics-model based approaches dominate all other baselines. In contrast to observation anomalies, action anomalies can only be observed implicitly or after some interaction. This requires modeling environment dynamics explicitly which again highlights the importance of differentiating AD in RL from other related areas, as the aspects that are specific to RL (actions, dynamics) play a critical role.

**Analysis on Anomaly Strengths.** Figure 2 c) and f) show the influence of anomaly strength on detection performance. In general we observe that detection performance strongly correlates with anomaly strength. Interestingly, even the best performing detectors exhibit high variance for tiny anomalies. For strong anomalies, this spread is significantly smaller. This demonstrates that the calibration on policy degradation results in meaningful difficulty levels, rather than idiosyncratic parameter choices as done in previous works.

**Influence of Rollout Policy.** To analyze the influence of the rollout policy on anomaly detection performance, we conducted additional experiments with different RL policies. We trained two additional policies (SAC, TD3) for 3 different environments until they reach a comparable success rate (normalized score $\approx 1$). We then created datasets and evaluated detectors on these new datasets (strength levels remain those tuned with TQC). Table 3 a) shows that RL policies with similar success rates in nominal environments respond differently to anomalies, with TQC being the most robust. Tables Table 3 a) & b) present detection performance (AUROC) of the three best detectors for vec./img observations, showing that while results depend on the rollout policy, the relative ranking of detection performance remains consistent.

**Analyisis of Detection Timing.** Beyond aggregated detection metrics such as AUROC, the timely detection of anomalies is important. To showcase how our benchmark can be utilized to study detection times, we plot detection-delay distributions across thresholds and detectors. We select thresholds exclusively from a small normal-only validation split using three common rules with different operating characteristics (see Figure 3 and details in Appendix A.8). We define detection delay as $\Delta t = t^* - t_a$ where $t_a$ is the ground-truth of anomaly onset and $t^*$ the earliest time a detector identifies an anomaly. Generally, detectors for both vector and image observations show long tails in the distribution of detection times, highlighting the difficulty of threshold selection. We also see again, that AD from images is significantly harder, with wider, more off-centered distributions. More detailed results, grouped by environment domain are available in Appendix A.10.

**Sim-to-real alignment**: Trends observed in the real-world URRtde-Reach task qualitatively match its simulated counterpart URM-Reach (see Appendix A.9). While verification on more real-world environments is needed to show the general applicability of our findings, this is a promising finding, suggesting that insights from Anomaly-Gym transfer beyond simulators.

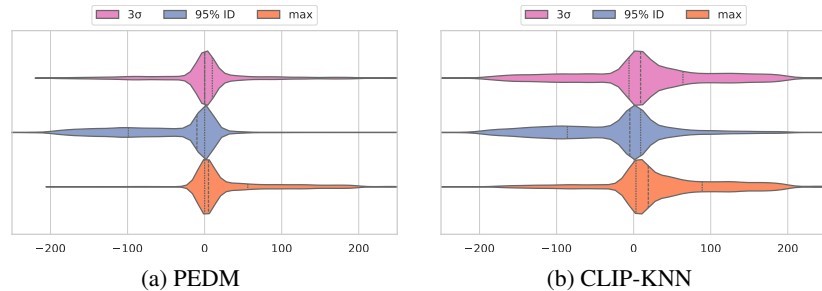

**Figure 3**: Timing Results: Distribution of detection delays. Dotted lines: $25^{th}/75^{th}$ percentile, dashed line: $50^{th}$ percentile. Positive indicates delayed detection, negative values early detection.

## 7 CONCLUSION, LIMITATIONS, AND OUTLOOKS

In this work, we introduced Anomaly-Gym, the first large-scale benchmark for AD in RL. Anomaly-Gym offers a diverse suite of environments, anomaly types, strength levels, observation modalities, and data from both simulated and real-world tasks. In a series of experiments, we analyzed the importance of these different aspects and compared various detection methods. Our analysis reveals the significant impact of anomaly strength and type on detection performance and underscores the open challenges posed by image-based observations, threshold selection and timely detection.

**Limitations (of our evaluations)**
*(i)* We do not consider cross-policy transfer settings, where a detector is trained on data from one policy, and evaluated on data from another policy. This a deliberate choice reflecting realistic deployments. Nonetheless, this setting is possible with Anomaly-Gym.
*(ii)* We focus on radom onset schemes where anomalies persist until the episode ends. Single-point and group events with recovery require further study. Anomaly-Gym already supports these features. However, not all environments might allow recovery after failure, causing potential label ambiguity.
*(iii)* Due to a lack of established methods for image observations, we only compared a limited number of and baselines. Future research should expand these evaluations with a broader range of methods from related fields. Our contribution is the benchmark itself, not the detection methods. Our results however suggest concrete directions for future work: The combination of sophisticated encoder architectures (e.g. ViTs) with architectures that explicitly model environment dynamics.
*(iv)* We treat anomaly detection and robustness as disentangled. However, many real-world applications demand both robustness to minor perturbations and the ability to detect severe anomalies. An important direction for future research is therefore the joint study of these two problems, including how to determine detection thresholds from normal data while accounting for policy robustness. Our work already offers the necessary tools to explore these questions.
*(v)* While the framework we introduce in this work is policy agnostic, we focus on data induced by RL policies. Deep RL's reliance on neural networks requires external monitoring, as white-box techniques are not available. This makes AD especially relevant for RL policies.

**Outlook (future benchmark extensions)**
*(i)* Anomaly-Gym currently only includes one real-world environment. Although the results from this real-world environment are comparable with the simulated data, verification in a broader variety of real-world applications remains crucial. However, collecting anomalous real-world data is inherently difficult because anomalies are rare and often unsafe to induce: in robotics they risk hardware damage, and in healthcare they can be unethical due to potential harm to subjects.
*(ii)* We currently omit semantic visual shifts (e.g. weather, novel objects). Such shifts are highly environment specific and often irrelevant to the policy (e.g. background objects). This limits tunability and cross-task comparability, and potentially adds label ambiguity (is a background-only shift an anomaly?). We believe this topic deserves a dedicated study which we leave to future work. Anomaly-Gym allows easy implementation of such anomalies.

Ultimately, Anomaly-Gym provides an important foundation that enables researchers to systematically evaluate and compare novel AD methods and drive the development of robust and reliable RL agents for real-world applications.

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

# A APPENDIX

## A.1 DESCRIPTIONS OF ENVIRONMENTS IN ANOMALY-GYM

See Figure 1 for a visualization of all envs. Table 4 provides a detailed overview of all environments. For even more detail, we refer to the implementation available at: `https://anonymous.4open.science/r/iclr-18811`.

Table 4: List of Environments

| Env_id | Description | ObservationTypes/Spaces | Action-Space, Description | Anomaly Types |
|---|---|---|---|---|
| Carla-LaneKeep | Follow the lane at target speed and don't collide with other vehicles | - vector: (9,)
– current speed
– target speed
– current accell.
– current heading
– dist. to lane center
– dist to veh. ahead
– delta vel. to veh. ahead
– last accel.
– last steering angle
- img: (3x256x256) | (2,), acceleration and steering angle | - brake fail
- steer fail
- slippery road
- Action Factor/Offset/Noise/ Delay/Temp. Noise/Drift
- Observation Factor/Offset/Noise/Temp. Noise/Quantization/Drift/ |
| Mujoco-CartpoleSwingup | Swingup the Pole by moving the cart | - vector: (4,)
– car pos.
– cart vel.
– pole angle
– pole vel.
- img: (3x128x128) | (1,) move the cart left/right | - Mass Factor
- Force Vector
- joint friction
- Action Factor/Offset/Noise/ Delay/Temp. Noise/Drift
- Observation Factor/Offset/Noise/Temp. Noise/Quantization/Drift/ |
| Mujoco-Reacher3D | Move Robot EE to goal | - vector: (17,)
– joint pos.
– joint vel.
- img: (3x128x128) | (7,) Torque applied on the robot joints | - Mass Factor
- Force Vector
- joint friction
- Action Factor/Offset/Noise/ Delay/Temp. Noise/Drift
- Observation Factor/Offset/Noise/Temp. Noise/Quantization/Drift/ |
| Mujoco-HalfCheetah | Control the HalfCheetah to move as fast as possible | - vector: (18,)
– linear vel.
– joint pos.
– joint vel.
- img: (3x128x128) | (6,) Torque applied on the robot joints | - Mass Factor
- Force Vector
- joint friction
- Action Factor/Offset/Noise/ Delay/Temp. Noise/Drift
- Observation Factor/Offset/Noise/Temp. Noise/Quantization/Drift/ |
| Sape-Goal0 | Move to goal while avoiding collisions with obstacle between agent and goal | - vector: (26,)
– agent pos
– agent vel
– goal pos
– object lidar
– hazard lidar
- img: (3x128x128) | (2,) acceleration in x-/y direction | - Force Agent
- Moving Objects
- Moving Friction
- Action Factor/Offset/Noise/ Delay/Temp. Noise/Drift
- Observation Factor/Offset/Noise/Temp. Noise/Quantization/Drift/ |
| Sape-Goal1 | Move to goal while avoiding collisions with multiple obstacles & hazards spwaned around the goal | - vector: (26,)
– agent pos
– agent vel
– goal pos
– object lidar
– hazard lidar
- img: (3x128x128) | (2,) acceleration in x-/y direction | - Force Agent
- Moving Objects
- Moving Friction
- Action Factor/Offset/Noise/ Delay/Temp. Noise/Drift
- Observation Factor/Offset/Noise/Temp. Noise/Quantization/Drift/ |
| Sape-Goal2 | Move to goal while avoiding collisions with multiple obstacles & hazards spwaned around the goal | - vector: (26,)
– agent pos
– agent vel
– goal pos
– object lidar
– hazard lidar
- img: (3x128x128) | (2,) acceleration in x-/y direction | - Force Agent
- Moving Objects
- Moving Friction
- Action Factor/Offset/Noise/ Delay/Temp. Noise/Drift
- Observation Factor/Offset/Noise/Temp. Noise/Quantization/Drift/ |
| URMujoco-Reach | Move end-effector to target | - vector: (13,)
– ee pos
– ee orientation (rpy)
–goal pos
–goal orientation (quat)
– img: (3x128x128) | (3,) displacement of robot end-effector in cartesian space | - Robot Speed
- Moving Goal
- Robot Friction
- Action Factor/Offset/Noise/ Delay/Temp. Noise/Drift
- Observation Factor/Offset/Noise/Temp. Noise/Quantization/Drift/ |
| URMujoco-PnP | Pick up box and move to target | - vector: (13,)
– ee pos
– ee orientation (rpy)
–goal pos
–goal orientation (quat)
–gipper state
–block position
–block orientation
– img: (3x128x128) | (4,) displacement of robot end-effector in cartesian space, gripper distance target | - Robot Speed
- Moving Goal
- Robot Friction
- Action Factor/Offset/Noise/ Delay/Temp. Noise/Drift
- Observation Factor/Offset/Noise/Temp. Noise/Quantization/Drift/ |
| URRtde-Reach | Move end-effector to target | - vector: (13,)
– ee pos
– ee orientation (rpy)
–goal pos
–goal orientation (quat)
– img: (3x256x256) | (3,) displacement of robot end-effector in cartesian space | - Control Latency
- Moving Goal
- Control Smoothing
- Action Factor/Offset/Noise/ Delay/Temp. Noise/Drift
- Observation Factor/Offset/Temp. Noise/Quantization/Drift/ |

## A.2 DESCRIPTIONS OF ANOMALIES

In the following, we describe all anomalies in more detail.

**Observation Anomalies** are perturbations to the observations emitted by the environment. We implement the following observation anomalies:

- Noise: $o'_t = o_t + e_t$
- Scaling: $o'_t = o_t \cdot \beta$
- Offset: $o'_t = o_t + \beta$
- Drift: $o'_t = o_t + \beta * t$
- Temporal Noise: $o'_t = o_t + \beta\, n_{t-1} + \varepsilon_t +$
- Quantization: $o'_t = \beta \cdot \left\lfloor \frac{o_t}{\beta} \right\rceil$

where $\quad \varepsilon_t \sim \mathcal{N}(0, \sigma^2), \quad n_0 = \varepsilon_0$ and $\beta \in \mathbb{R}^+$

**Action Anomalies** are perturbations to the actions before they are applied to the environment. We implemented the same types of anomalies as those used for observations but instead of quantization, we add action delay: $a'_t = a_{t-\beta}$ where $\beta \in \mathbb{N}^+$.

**Dynamics Anomalies** are perturbations to the dynamics operator $\mathcal{T} : \mathcal{S} \times \mathcal{A} \to \mathcal{S}$ of the environment. We implement the following dynamics anomalies.

- Body Mass: Body mass is multiplied by a constant factor
- Force Vector: Constant force vector applied to the center of a single robot joint (MJC, URM) or to the center of Agent (SAP)
- Friction: of robot joints changed
- Damping: Inertia of agent is reduced by a factor $\beta$ in each timestep
- Moving Object: Objects moving in uniformly random directions with increasing speed
- Brake Fail: Braking force reduced
- Steer Fail: Steering effect reduced
- Slippery Road: Friction parameters of the wheel to the road surface are reduced
- Moving Goal: Moving Goal back and forth on a straight line
- Control Smoothing: Increased control smoothing of low-level Robot Controllers
- Control Latency: Increased latency of low-level Robot Controllers
- Robot Speed: Max moving speed of robot joints reduced

For more details, we refer to the implementation of each environment's anomalies.

### A.3 SATISFACTION OF DESIDERATA

### A.4 DETAILS ON ANOMALY STRENGTH TUNING

As outlined in Section 5.3 tune the magnitude of all anomalies and define different strength levels w.r.t. the degree of degradation of the rollout policy. Let

$$J_{\mathcal{M}}(\pi) = \mathbb{E}\left[\sum_{t=0}^{T} r_t\right]$$

be the average cumulative reward of some policy $\pi$ on some MDP $\mathcal{M}$, where $r_t$ is the reward received at time step $t$ and $T$ is the time horizon. Let $J_{\mathcal{M}^+}(\pi_\theta)$, be the average cumulative reward of the trained policy, $J_{\mathcal{M}^+}(\pi_R)$ be the average cumulative reward of a random policy, and $J_{\mathcal{M}^-}(\pi_\theta)$ be the average cumulative reward of the trained policy on the anomalous MDP.

We define policy degradation via the normalized score:

$$\bar{J}_{\mathcal{M}^-}(\pi_\theta) = \frac{J_{\mathcal{M}^-}(\pi_\theta) - J_{\mathcal{M}^+}(\pi_R)}{J_{\mathcal{M}^+}(\pi_\theta) - J_{\mathcal{M}^+}(\pi_R)} \tag{5}$$

and set the different levels of anomaly strength at

- *tiny*: $\bar{J}_{\mathcal{M}^-}(\pi_\theta) \approx 0.99$,
- *medium*: $\bar{J}_{\mathcal{M}^-}(\pi_\theta) \approx 0.90$
- *strong*: $\bar{J}_{\mathcal{M}^-}(\pi_\theta) \approx 0.75$
- *extreme*: $\bar{J}_{\mathcal{M}^-}(\pi_\theta) \approx 0.50$

Table 5: Satisfaction of Desiderata in anomaly-gym

| desiderata | Satisfaction |
|---|---|
| **ED1-Diversity.** | Anomaly-Gym includes 10 different environments from 4 different fields: Autonomous Driving(Carla), Robotics Manipulation (UR-Envs), Robotics Control (Mujoco-Control) and navigation (Sape). |
| **ED2-Scalability.** | Environment observation sizes range from $\mathcal{O} \in \mathbb{R}^4$ to $\mathcal{O} \in \mathbb{R}^{26}$ for vector observations or $\mathbb{R}^{3x256x256}$ for image observations. Action spaces range from $\mathcal{A} \in \mathbb{R}^2$ to $\mathcal{A} \in \mathbb{R}^7$ respectively. |
| **ED3-Realism.** | URRtde is a real world Environment. |
| **ED4-Solvability.** | We train a policy until a success rate of ¿90 % is reached in each environment. Sucess is defined as follows. Sape-Goal{1,2,3}: Reach the goal without collision. Carla-LaneKeep: target speed without collision at end of episode. URM-PnP: box at target. Mujoco-Reacher3d, URMujoco-Reach, URRted-Reach: End-effector at target. Cartpole-Swingup: End-effector displacement smaller than trheshold. Mujoco-HalfCheetah: Final ground-speed larger than threshold. |
| **ED5-Reproducibility.** | Initial conditions are controlled by consitend random seeding. |
| **ED6-Configurability.** | All environment parameters can be adjusted in all be the real-world environment. |
| **AD1-Diversity.** | Anomaly-Gym covers a range of 19 different anomalies. |
| **AD2-Realism.** | Observation anomalies can occur in real world in the form of sensor variations or failure. Action anomalies can occur in the real world in form of actuator variations or failure. Dynamics anomalies can occur in the real world as external influences on the environment/agent. |
| **AD3-Impact.** | We show that all but two anomaly types (moving objects, moving goal) can decrease agent normalized scores by as least as much as 50%. |
| **AD4-Difficulty.** | We exclude anomalies such as sensor/actuator shutdown or failure. All anomalies (especially tiny) minimally alter the original MDP. |

To do this, we ran a fine grid search for each anomaly, strength and environment combination. Since this is a noisy process, we used 500 episodes for each sample. For most anomaly types, there is a strong correlation between anomaly strength and policy performance. For some anomalies, however, increasing the strength parameter does not lead to any further degradation of the policy after some point. This is exemplified in Figure 4, for three different anomaly types in the Sape-Goal1 environment. A force on the agent, as well as an offset of the action, leads to a decrease in policy performance with increasing anomaly strength. Moving objects with higher speeds, on the other hand, do not continue to influence policy performance after a certain point. The same behavior can be observed for all moving-object anomalies in SAP and for the moving-goal anomaly in URM-PnP. We thus exclude Sape-Goal-1,2,3 moving objects (strong, extreme) and URM-PnP moving goal (extreme) during our evaluation. We also had to omit noise and temp-noise anomalies for URRtde-Reach due to potential hardware damage caused by oscillating control commands.

Nonetheless, this process allows us to compare all different types of anomalies along the same strength levels and to compare the same anomaly types across different environments.

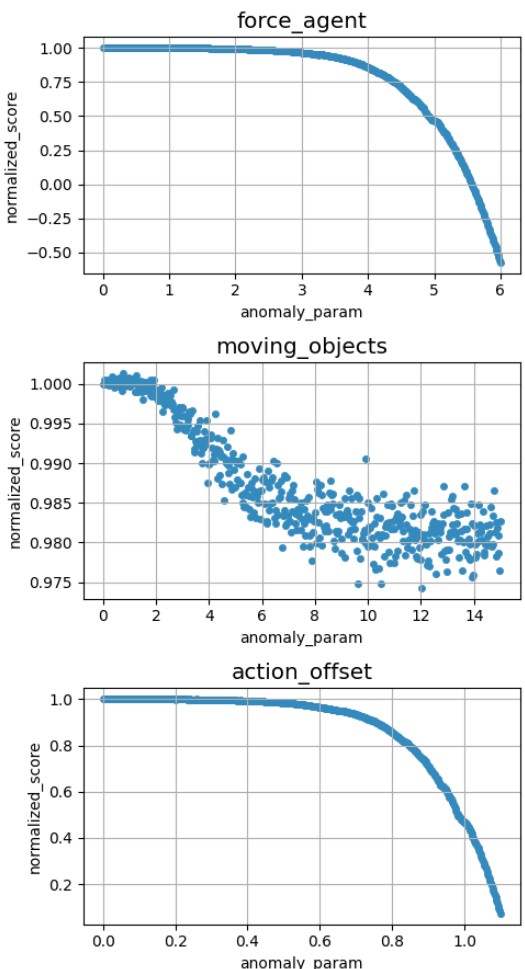

Figure 4: Example of anomaly strength tuning process in Sape-Goal1.

### A.5 EXACT ANOMALY PARAMETERS

See Table 7

### A.6 DETAILS ON AGENT TRAINING

We train a TQC agent Kuznetsov et al. (2020) for each base environment, using an implementation from Raffin et al. (2021) and hyper-parameters presented in Table 8. Hyper-parameters were selected with a TPE (Tree-structured Parzen Estimator) sweep over learning rate, replay-buffer-size, batch-size, network size, gamma and lr-schedule.

### A.7 DETAILS ON DETECTION MODELS

All detectors return anomaly labels:

$$D(x) = \begin{cases} 1 & \text{if } \psi(\cdot) > \vartheta \\ 0 & \text{otherwise} \end{cases}, \tag{6}$$

where 1 represents anomalous samples, and 0 normal samples. The score function and additional hyper-parameters are displayed in Table 8. Hyper-parameters were selected with a coarse grid search over the first 5 parameters for each method in this table.

Table 7: Exact Anomaly Parameters

| | anomaly | medium | strong | tiny | extreme |
|---|---|---|---|---|---|
| Carla-LaneKeep | action scaling | 2.91 | 4.18 | 1.73 | 5.91 |
| | action noise | 0.86 | 1.04 | 0.31 | 1.41 |
| | action offset | 0.55 | 0.65 | 0.35 | 0.86 |
| | brake fail | 0.11 | 0.08 | 0.40 | 0.04 |
| | observation scaling | 1.41 | 2.06 | 1.00 | 3.61 |
| | observation noise | 0.09 | 0.17 | 0.01 | 0.32 |
| | observation offset | 0.04 | 0.05 | 0.01 | 0.31 |
| | slippery road | 0.11 | 0.09 | 0.22 | 0.06 |
| | steer fail | 0.24 | 0.10 | 0.32 | 0.05 |
| Mujoco-CartpoleSwingup | action scaling | 2.19 | 3.21 | 1.38 | 4.58 |
| | action noise | 1.11 | 1.44 | 0.40 | 1.74 |
| | action offset | 0.82 | 0.90 | 0.49 | 0.96 |
| | observation scaling | 1.03 | 1.04 | 1.01 | 1.05 |
| | observation noise | 0.09 | 0.11 | 0.02 | 0.15 |
| | observation offset | 0.06 | 0.09 | 0.02 | 0.14 |
| | robot force | 9.77 | 14.58 | 3.46 | 21.54 |
| | robot friction | 1.58 | 2.00 | 0.60 | 2.18 |
| | robot mass | 1.63 | 2.72 | 1.26 | 3.29 |
| Mujoco-HalfCheetah | action scaling | 1.16 | 1.26 | 1.09 | 1.59 |
| | action noise | 0.09 | 0.15 | 0.03 | 0.27 |
| | action offset | 0.08 | 0.11 | 0.03 | 0.16 |
| | observation scaling | 1.08 | 1.14 | 1.03 | 1.31 |
| | observation noise | 0.01 | 0.01 | 0.00 | 0.03 |
| | observation offset | 0.01 | 0.01 | 0.00 | 0.04 |
| | robot force | 10.38 | 22.44 | 1.19 | 32.55 |
| | robot friction | 4.21 | 8.96 | 0.48 | 14.79 |
| | robot mass | 1.13 | 1.21 | 1.02 | 1.40 |
| Mujoco-Reacher3D | action scaling | 2.28 | 2.50 | 1.36 | 2.50 |
| | action noise | 0.69 | 1.18 | 0.19 | 1.80 |
| | action offset | 0.41 | 0.65 | 0.10 | 0.82 |
| | observation scaling | 1.35 | 1.70 | 1.10 | 2.51 |
| | observation noise | 0.44 | 1.49 | 0.07 | 5.00 |
| | observation offset | 0.20 | 0.41 | 0.06 | 0.62 |
| | robot force | 7.27 | 10.00 | 3.15 | 10.00 |
| | robot friction | 0.40 | 0.62 | 0.10 | 0.82 |
| | robot mass | 9.60 | 33.48 | 1.74 | 75.00 |
| Sape-Goal0 | action scaling | 0.22 | 0.11 | 0.62 | 0.03 |
| | action noise | 1.02 | 2.00 | 0.29 | 4.50 |
| | action offset | 0.69 | 0.89 | 0.12 | 1.00 |
| | force agent | 3.45 | 4.46 | 0.60 | 5.01 |
| | mass agent | 4.46 | 8.92 | 1.61 | 20.00 |
| | moving objects | 4.87 | - | 0.91 | - |
| | observation scaling | 2.57 | 4.08 | 1.41 | 9.93 |
| | observation noise | 0.47 | 0.84 | 0.04 | 1.53 |
| | observation offset | 0.30 | 0.40 | 0.10 | 0.47 |
| Sape-Goal1 | action scaling | 0.22 | 0.11 | 0.55 | 0.03 |
| | action noise | 1.55 | 2.40 | 0.67 | 5.00 |
| | action offset | 0.78 | 0.88 | 0.50 | 0.99 |
| | force agent | 3.88 | 4.42 | 2.49 | 4.97 |
| | mass agent | 4.50 | 8.92 | 1.80 | 19.92 |
| | moving objects | 13.44 | - | 3.47 | - |
| | observation scaling | 3.22 | 4.88 | 1.63 | 10.00 |
| | observation noise | 0.51 | 0.79 | 0.22 | 1.53 |
| | observation offset | 0.26 | 0.36 | 0.13 | 0.43 |
| Sape-Goal2 | action scaling | 0.21 | 0.10 | 0.55 | 0.01 |
| | action noise | 1.22 | 1.89 | 0.54 | 3.59 |
| | action offset | 0.67 | 0.84 | 0.37 | 0.95 |
| | force agent | 3.35 | 4.19 | 1.85 | 4.74 |
| | mass agent | 4.69 | 9.61 | 1.91 | 20.00 |
| | moving objects | 34.07 | - | 3.01 | - |
| | observation scaling | 1.61 | 2.43 | 1.16 | 9.95 |
| | observation noise | 0.29 | 0.45 | 0.10 | 0.74 |
| | observation offset | 0.14 | 0.24 | 0.06 | 0.60 |
| URMujoco-Reach | action factor | 0.915 | 0.795 | 0.590 | 0.301 |
| | action noise | 0.376 | 0.667 | 0.929 | 1.341 |
| | action offset | 0.295 | 0.501 | 0.657 | 0.794 |
| | moving goal | 0.003 | 0.004 | 0.004 | 0.004 |
| | obs factor | 1.032 | 1.087 | 1.095 | 1.107 |
| | obs noise | 0.002 | 0.006 | 0.016 | 0.026 |
| | obs offset | 0.018 | 0.023 | 0.024 | 0.026 |
| | robot friction | 13.026 | 80.160 | 190.381 | 468.938 |
| | robot speed | 0.009 | 0.003 | 0.003 | 0.002 |
| URRtde-Reach | action factor | 0.935 | 0.782 | 0.596 | 0.313 |
| | action noise | 0.263 | 0.566 | 0.990 | 1.394 |
| | action offset | 0.364 | 0.545 | 0.687 | 0.788 |
| | control smoothing | 0.061 | 0.077 | 0.104 | 0.173 |
| | control latency | 0.007 | 0.020 | 0.042 | 0.098 |
| | moving goal | 0.001 | 0.003 | 0.003 | 0.004 |
| | obs factor | 1.039 | 1.082 | 1.094 | 1.103 |
| | obs noise | 0.002 | 0.008 | 0.016 | 0.024 |
| | obs offset | 0.016 | 0.023 | 0.025 | 0.025 |
| URMujoco-PnP | action factor | 0.871 | 0.709 | 0.491 | 0.305 |
| | action noise | 0.295 | 0.402 | 0.492 | 0.742 |
| | action offset | 0.091 | 0.242 | 0.447 | 0.561 |
| | moving goal | 0.003 | 0.006 | 0.042 | - |
| | obs factor | 1.227 | 1.439 | 1.712 | 2.091 |
| | obs noise | 0.002 | 0.005 | 0.008 | 0.011 |
| | obs offset | 0.006 | 0.012 | 0.017 | 0.022 |
| | robot friction | 0.000 | 8.838 | 46.717 | 121.212 |
| | robot speed | 0.004 | 0.003 | 0.002 | 0.001 |

## A.8 THRESHOLD SELECTION

While overall discriminative performance is an important measure of effective anomaly detectors, the timeliness of detection is also a critical aspect. To quantify this, we analyze the detection delay. We define detection delay as:

$$\Delta t = t^* - t_a \tag{7}$$

where $t_a$ is the ground-truth of anomaly onset and $t^*$ the earliest time a detector identifies an anomaly. A positive $\Delta t$ indicates delayed detection (implies false negatives); a negative value means early detection (implies false positives).

For this, detection methods require a fixed operating point, i.e., a specific threshold, beyond which samples are identified as anomalous. In the absence of labeled anomalies, a common approach is to fit a model to the normal data and select a threshold based on the distribution of anomaly scores on normal inputs of a validation set Aggarwal (2016):

1. Three-sigma rule: $\vartheta = \mu + 3\sigma$, where $\mu$ and $\sigma$ are the mean and standard deviation of scores under normal data

2. Quantile threshold: $\vartheta = Q_{0.95}(d(x)|x \in \mathcal{D}_{\text{train}})$, the 95th percentile of anomaly scores on normal data.

3. Max-validation threshold: $\vartheta = \max_{x \in \mathcal{D}_{\text{train}}} d(x)$, the maximum anomaly score on normal data.

Table 9: Details on detection-model parameters and score functions

| model | parameters | score function |
|---|---|---|
| IF | n-estimators: 100 | $\psi(s) = -2^{-\frac{E(h(s))}{2 \cdot \ln(n-1) + \gamma}}$ |
| KNN | k: 1 | $\psi(s) = \|s - x_{k=1}\|_2$ |
| OCSVM | kernel: 'rbf'
degree: 3
gamma: 'scale'
coef: 0.0
tol: 0.001m
nu: 0.5 | $\psi(x) = \sum_{i=1}^{N} \alpha_i K(x, x_i) - \rho$ |
| riqn | gru-units: 64
quantile-embedding-dim: 128
num-quantile-sample: 64
num-tau-sample: 1
lr: 0.001
train-epochs: 250 | $\psi(s, s') = \|f(s) - s'\|$ |
| MLP-DM | network-size: 512-256-128
weight-decay: 0.0001
lr: 0.001
train-epochs: 250 | $\psi(s, a, s') = \|f(s, a) - s'\|_2$ |
| LSTM-DM | hidden-dim: 256
num-layers: 1
fully-connected-dim: 128
weight-decay: 0.0001
lr: 0.001
train-epochs: 250 | $\psi(s, a, s') = \|f(s, a) - s'\|_2$ |
| PE-DM | network-size: 512-256-128
weight-decay: 0.0001
ens-size: 5
n-samples: 1000
lr: 0.001
train-epochs=250 | $\psi(s, a, s') = \frac{1}{B}\{\|f(s, a)^b - s'\|_2\}^B$ |

| AE | channel-sizes: 32-64-128-256
feature-size: 128
lr: 0.0005
train-epochs: 250 | $\psi(o) = \|f(o) - o\|_2$ |
|---|---|---|
| PredAE | channel-sizes: 32-64-128-256
feature-size: 128
lr: 0.0001
train-epochs: 250 | $\psi(o, o') = \|f(o) - o'\|_2$ |
| KNN-AE | channel-sizes: 32-64-128-256
feature-size: 128
lr: 0.0005
train-epochs: 250 | $\psi(o, o') = \|f(o) - f(o_{k=1})\|_2$ |
| KNN-ResNet | model name: Resnet18
k:1 | $\psi(o, o') = \|f(o) - f(o_{k=1})\|_2$ |
| PredNet | A-channels: (3, 48, 96, 192)
R-channels: (3, 48, 96, 192)
num-layers: 3
nt: 10
lr: 0.001
train-epochs: 100 | $\psi(o_{[t-10:t]}, o') = \|f(o_{[t-10:t]}) - o'\|_2$ |
| LDM | channel-sizes: 32-64-128-256
feature-size: 128
lr: 0.0005
train-epochs: 250 | $\psi(o, a, o') = \|f(o, a) - o'\|_2$ |
| DINO-Patch | model name: vit-patch16-224 | $\psi(o) = \max_{p \in \mathcal{P}(o)} \min_{p' \in \mathcal{P}(\mathcal{D})} \|f(p) - f(p')\|_2$ |
| CLIP-Knn | model name = ViT-B/32 | $\psi(o, o') = \|f(o) - f(o_{k=1})\|_2$ |

### A.9 REAL WORLD EXPERIMENTS

To validate empirical findings with simulated data, Anomaly-Gym also includes one real-world environment. *URRtde-Reach* is in its core a replica of its simulated version *URM-Reach*. Instead of relying on the MuJoCo physics simulator Todorov et al. (2012), *URRtde-Reach* employs a real-time *RTDE*Lindvig et al. (2025) interface to a physical UR3CB robotic manipulator, as well as an Intel RealSense camera interface for obtaining image observations. Apart from these interfaces, both environments are identical. We can thus compare both environments on a 1-to-1 basis.

To gather data with *URRtde-Reach*, we use a policy trained in MuJoco and apply it zero-shot to the real-world task. Since real-world interaction is time-consuming and expensive, we only collect 50 episodes for each train/test/val/anomaly setting instead of 100. Figure 5 compares the distribution of AUROC scores of different detectors in both environments. Although there is a visible difference, the general trend and order stay the same. More fine-grained results are provided in Appendix A.12

### A.10 DETECTION TIMING RESULTS PER ENVIRONMENT

See Figure 6.

### A.11 COMPUTE RESOURCES

We performed all but one of our experiments on a single node of a local compute cluster with the following configuration:

- GPU NVIDIA L40 (46068MiB) - RAM 64GB - CPU AMD EPYC 9334 32-Core

All experiments with vector observations consumed <5GB of GPU Memory peak. Experiments with image observations were more memory-intensive. GPU Memory consumption was, however, still moderate (<10GB), apart from one exception: Experiments with Carla-LaneKeep, which, due to its long episode length, required between 5 GB for Knn-ResNet (the most memory efficient

1080
1081
1082
1083
1084
1085
1086
1087
1088
1089
1090
1091
1092
1093
1094
1095
1096
1097
1098
1099

Table 8: Hyper parameters

| Parameter | CAR/SAP | MJC | UR-* |
|---|---|---|---|
| learning rate | 0.0003 | 0.001 | 0.005 |
| replay buffer size | 1e6 | 1e6 | 1e6 |
| learning starts | 100 | 1e3 | 1e3 |
| batch size | 256 | 512 | 512 |
| tau | 0.005 | 0.005 | 0.005 |
| gamma | 0.99 | 0.98 | 0.98 |
| train freq | 1 | 1 | 1 |
| gradient steps | 1 | 1 | 1 |
| top quantiles to drop per net | 2 | 2 | 2 |
| Network size | 256-256 | 512-512-512 | 512-512-512 |
| n critics | 2 | 2 | 2 |
| lr schedule | none | linear anneal. | linear anneal. |

1100
1101
1102
1103
1104
1105
1106
1107
1108
1109
1110
1111
1112
1113
1114
1115
1116
1117
1118
1119
1120
1121
1122
1123
1124
1125
1126
1127
1128
1129
1130
1131
1132
1133

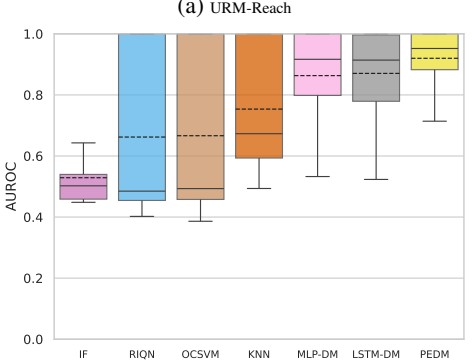
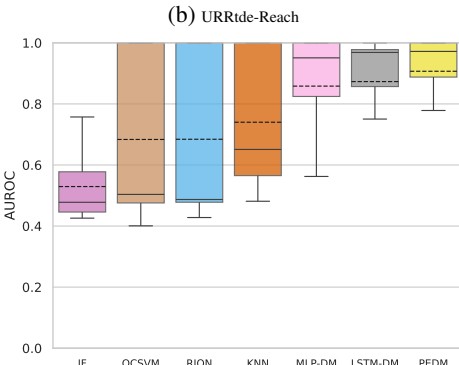

Figure 5: Simulated vs real-world data (vector observations)

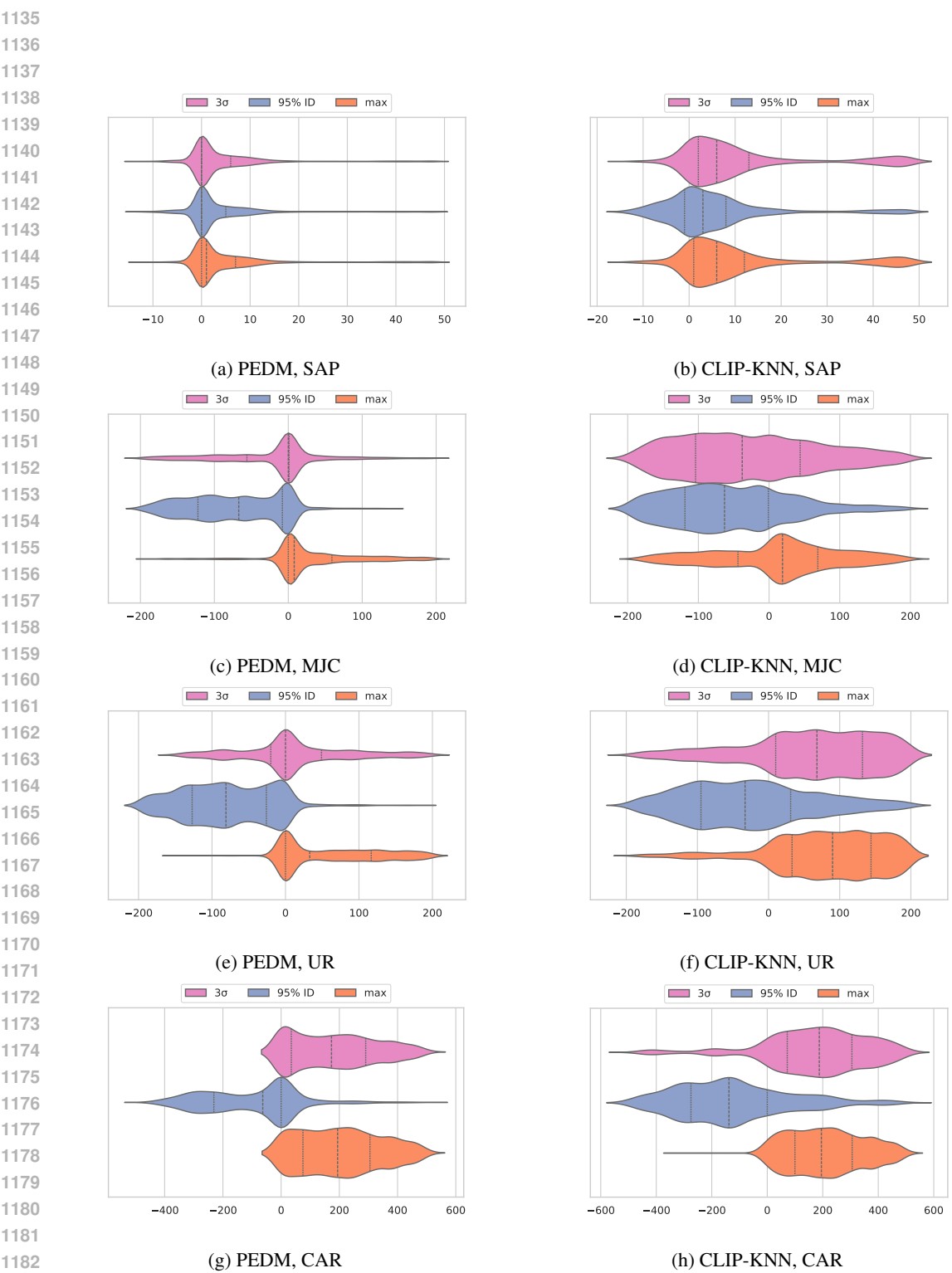

(a) PEDM, SAP      (b) CLIP-KNN, SAP

(c) PEDM, MJC      (d) CLIP-KNN, MJC

(e) PEDM, UR      (f) CLIP-KNN, UR

(g) PEDM, CAR      (h) CLIP-KNN, CAR

Figure 6: Detection Delays per environment domain for PEDM and CLIP-KNN

model) and 72GB for PredNet (the most memory extensive model). For the latter, we ran on an NVIDIA H100.

## A.12 DETAILED RESULTS

Table 10: Detailed results for all detectors and environments on vector observations. The mean and standard deviation are reported for each detector, environment, and anomaly strength. The results are grouped by environment and detector.

| env | strength | detector | AUROC | | AUPR | | FPR95 | | VUSPR | | VUSROC | | Global-AUROC | | Global-AUPR | |
|---|---|---|---|---|---|---|---|---|---|---|---|---|---|---|---|---|
| | | | mean | std | mean | std | mean | std | mean | std | mean | std | mean | std | mean | std |
| CAR-LaneKeep | strong | IF | 0.71 | 0.19 | 0.70 | 0.17 | 0.54 | 0.19 | 0.70 | 0.17 | 0.71 | 0.19 | 0.76 | 0.17 | 0.70 | 0.19 |
| | | KNN | 0.91 | 0.14 | 0.90 | 0.14 | 0.20 | 0.24 | 0.90 | 0.14 | 0.92 | 0.14 | 0.92 | 0.13 | 0.91 | 0.14 |
| | | LSTM-DM | 0.89 | 0.18 | 0.86 | 0.18 | 0.25 | 0.33 | 0.86 | 0.18 | 0.89 | 0.17 | 0.89 | 0.17 | 0.87 | 0.19 |
| | | MLP-DM | 0.89 | 0.17 | 0.85 | 0.17 | 0.26 | 0.31 | 0.85 | 0.17 | 0.89 | 0.17 | 0.90 | 0.16 | 0.87 | 0.17 |
| | | OCSVM | 0.77 | 0.20 | 0.75 | 0.19 | 0.46 | 0.26 | 0.75 | 0.19 | 0.77 | 0.19 | 0.81 | 0.18 | 0.77 | 0.20 |
| | | PEDM | 0.93 | 0.14 | 0.91 | 0.15 | 0.16 | 0.28 | 0.91 | 0.15 | 0.93 | 0.14 | 0.93 | 0.14 | 0.92 | 0.15 |
| | | RIQN | 0.72 | 0.21 | 0.72 | 0.19 | 0.54 | 0.26 | 0.72 | 0.18 | 0.73 | 0.21 | 0.76 | 0.19 | 0.73 | 0.21 |
| | tiny | IF | 0.41 | 0.09 | 0.50 | 0.06 | 0.77 | 0.07 | 0.50 | 0.06 | 0.43 | 0.09 | 0.49 | 0.08 | 0.45 | 0.06 |
| | | KNN | 0.75 | 0.19 | 0.73 | 0.16 | 0.46 | 0.28 | 0.73 | 0.16 | 0.75 | 0.19 | 0.76 | 0.18 | 0.71 | 0.18 |
| | | LSTM-DM | 0.71 | 0.24 | 0.69 | 0.19 | 0.55 | 0.35 | 0.69 | 0.19 | 0.72 | 0.24 | 0.72 | 0.24 | 0.69 | 0.22 |
| | | MLP-DM | 0.71 | 0.22 | 0.67 | 0.16 | 0.56 | 0.33 | 0.67 | 0.16 | 0.72 | 0.22 | 0.74 | 0.21 | 0.68 | 0.19 |
| | | OCSVM | 0.43 | 0.09 | 0.49 | 0.06 | 0.76 | 0.06 | 0.50 | 0.06 | 0.45 | 0.09 | 0.52 | 0.09 | 0.46 | 0.07 |
| | | PEDM | 0.82 | 0.21 | 0.80 | 0.19 | 0.41 | 0.37 | 0.80 | 0.19 | 0.82 | 0.21 | 0.82 | 0.21 | 0.80 | 0.22 |
| | | RIQN | 0.39 | 0.08 | 0.49 | 0.06 | 0.82 | 0.04 | 0.49 | 0.06 | 0.41 | 0.08 | 0.47 | 0.07 | 0.45 | 0.06 |
| MJC-CartpoleSwingup | strong | IF | 0.45 | 0.16 | 0.49 | 0.05 | 0.67 | 0.16 | 0.49 | 0.05 | 0.46 | 0.15 | 0.63 | 0.12 | 0.54 | 0.06 |
| | | KNN | 0.90 | 0.21 | 0.89 | 0.17 | 0.20 | 0.26 | 0.89 | 0.17 | 0.90 | 0.21 | 0.94 | 0.14 | 0.95 | 0.11 |
| | | LSTM-DM | 0.93 | 0.18 | 0.92 | 0.16 | 0.12 | 0.25 | 0.92 | 0.16 | 0.93 | 0.18 | 0.95 | 0.14 | 0.97 | 0.10 |
| | | MLP-DM | 0.91 | 0.21 | 0.91 | 0.18 | 0.15 | 0.26 | 0.91 | 0.18 | 0.91 | 0.21 | 0.94 | 0.16 | 0.95 | 0.12 |
| | | OCSVM | 0.50 | 0.14 | 0.51 | 0.08 | 0.66 | 0.16 | 0.52 | 0.08 | 0.52 | 0.13 | 0.67 | 0.11 | 0.60 | 0.08 |
| | | PEDM | 0.94 | 0.15 | 0.93 | 0.15 | 0.11 | 0.25 | 0.94 | 0.15 | 0.94 | 0.15 | 0.96 | 0.12 | 0.97 | 0.09 |
| | | RIQN | 0.49 | 0.12 | 0.49 | 0.06 | 0.78 | 0.17 | 0.50 | 0.05 | 0.51 | 0.12 | 0.65 | 0.10 | 0.56 | 0.06 |
| | tiny | IF | 0.30 | 0.14 | 0.43 | 0.02 | 0.77 | 0.13 | 0.44 | 0.02 | 0.32 | 0.13 | 0.47 | 0.13 | 0.45 | 0.04 |
| | | KNN | 0.77 | 0.31 | 0.78 | 0.20 | 0.39 | 0.30 | 0.78 | 0.20 | 0.77 | 0.30 | 0.84 | 0.24 | 0.84 | 0.19 |
| | | LSTM-DM | 0.83 | 0.28 | 0.85 | 0.22 | 0.23 | 0.34 | 0.85 | 0.22 | 0.84 | 0.28 | 0.87 | 0.24 | 0.89 | 0.20 |
| | | MLP-DM | 0.79 | 0.30 | 0.81 | 0.23 | 0.33 | 0.32 | 0.81 | 0.22 | 0.79 | 0.29 | 0.84 | 0.26 | 0.85 | 0.22 |
| | | OCSVM | 0.33 | 0.09 | 0.43 | 0.03 | 0.76 | 0.12 | 0.43 | 0.03 | 0.35 | 0.09 | 0.50 | 0.10 | 0.45 | 0.04 |
| | | PEDM | 0.87 | 0.25 | 0.88 | 0.21 | 0.22 | 0.39 | 0.88 | 0.21 | 0.88 | 0.24 | 0.90 | 0.21 | 0.92 | 0.18 |
| | | RIQN | 0.28 | 0.08 | 0.41 | 0.03 | 0.93 | 0.08 | 0.41 | 0.02 | 0.30 | 0.07 | 0.40 | 0.11 | 0.42 | 0.04 |
| MJC-HalfCheetah | strong | IF | 0.63 | 0.09 | 0.66 | 0.09 | 0.71 | 0.11 | 0.66 | 0.09 | 0.65 | 0.09 | 0.74 | 0.07 | 0.71 | 0.08 |
| | | KNN | 0.94 | 0.04 | 0.94 | 0.04 | 0.24 | 0.17 | 0.94 | 0.04 | 0.95 | 0.03 | 0.97 | 0.02 | 0.96 | 0.02 |
| | | LSTM-DM | 0.94 | 0.04 | 0.94 | 0.04 | 0.21 | 0.21 | 0.94 | 0.04 | 0.95 | 0.04 | 0.97 | 0.02 | 0.96 | 0.02 |
| | | MLP-DM | 0.94 | 0.05 | 0.94 | 0.04 | 0.22 | 0.23 | 0.94 | 0.04 | 0.95 | 0.04 | 0.97 | 0.03 | 0.96 | 0.02 |
| | | OCSVM | 0.61 | 0.13 | 0.68 | 0.10 | 0.77 | 0.12 | 0.68 | 0.10 | 0.64 | 0.12 | 0.68 | 0.13 | 0.71 | 0.11 |
| | | PEDM | 0.93 | 0.05 | 0.94 | 0.05 | 0.26 | 0.22 | 0.94 | 0.05 | 0.94 | 0.05 | 0.96 | 0.03 | 0.96 | 0.02 |
| | | RIQN | 0.56 | 0.11 | 0.63 | 0.09 | 0.82 | 0.08 | 0.63 | 0.08 | 0.59 | 0.10 | 0.59 | 0.11 | 0.63 | 0.10 |
| | tiny | IF | 0.29 | 0.12 | 0.45 | 0.04 | 0.89 | 0.09 | 0.45 | 0.04 | 0.32 | 0.12 | 0.43 | 0.12 | 0.46 | 0.05 |
| | | KNN | 0.69 | 0.15 | 0.71 | 0.13 | 0.54 | 0.17 | 0.72 | 0.13 | 0.71 | 0.15 | 0.80 | 0.12 | 0.75 | 0.12 |
| | | LSTM-DM | 0.73 | 0.17 | 0.74 | 0.14 | 0.52 | 0.24 | 0.74 | 0.14 | 0.75 | 0.17 | 0.82 | 0.14 | 0.77 | 0.13 |
| | | MLP-DM | 0.71 | 0.20 | 0.74 | 0.15 | 0.57 | 0.26 | 0.74 | 0.15 | 0.73 | 0.19 | 0.81 | 0.16 | 0.77 | 0.15 |
| | | OCSVM | 0.44 | 0.05 | 0.50 | 0.03 | 0.74 | 0.07 | 0.51 | 0.03 | 0.47 | 0.05 | 0.50 | 0.08 | 0.49 | 0.04 |
| | | PEDM | 0.65 | 0.20 | 0.69 | 0.16 | 0.60 | 0.22 | 0.70 | 0.16 | 0.67 | 0.19 | 0.76 | 0.16 | 0.73 | 0.15 |
| | | RIQN | 0.50 | 0.04 | 0.54 | 0.03 | 0.75 | 0.02 | 0.55 | 0.03 | 0.54 | 0.04 | 0.54 | 0.04 | 0.53 | 0.04 |
| MJC-Reacher3D | strong | IF | 0.62 | 0.28 | 0.63 | 0.24 | 0.57 | 0.32 | 0.63 | 0.24 | 0.63 | 0.28 | 0.69 | 0.25 | 0.67 | 0.25 |
| | | KNN | 0.96 | 0.05 | 0.93 | 0.08 | 0.11 | 0.11 | 0.93 | 0.08 | 0.96 | 0.05 | 0.91 | 0.09 | 0.89 | 0.14 |
| | | LSTM-DM | 0.98 | 0.04 | 0.95 | 0.07 | 0.05 | 0.10 | 0.95 | 0.07 | 0.98 | 0.04 | 0.99 | 0.01 | 0.99 | 0.01 |
| | | MLP-DM | 0.96 | 0.07 | 0.92 | 0.10 | 0.09 | 0.15 | 0.92 | 0.10 | 0.96 | 0.07 | 0.98 | 0.03 | 0.97 | 0.03 |
| | | OCSVM | 0.71 | 0.27 | 0.69 | 0.26 | 0.45 | 0.33 | 0.69 | 0.26 | 0.72 | 0.26 | 0.73 | 0.26 | 0.74 | 0.26 |
| | | PEDM | 0.98 | 0.04 | 0.96 | 0.07 | 0.04 | 0.10 | 0.96 | 0.07 | 0.98 | 0.04 | 0.99 | 0.01 | 0.99 | 0.01 |
| | | RIQN | 0.56 | 0.30 | 0.57 | 0.23 | 0.61 | 0.32 | 0.57 | 0.23 | 0.57 | 0.29 | 0.62 | 0.28 | 0.62 | 0.26 |
| | tiny | IF | 0.28 | 0.09 | 0.38 | 0.03 | 0.83 | 0.08 | 0.39 | 0.03 | 0.30 | 0.09 | 0.37 | 0.09 | 0.37 | 0.04 |
| | | KNN | 0.88 | 0.08 | 0.82 | 0.11 | 0.23 | 0.13 | 0.83 | 0.11 | 0.88 | 0.08 | 0.69 | 0.15 | 0.61 | 0.15 |
| | | LSTM-DM | 0.87 | 0.17 | 0.81 | 0.17 | 0.25 | 0.31 | 0.81 | 0.17 | 0.87 | 0.17 | 0.90 | 0.14 | 0.85 | 0.15 |
| | | MLP-DM | 0.73 | 0.27 | 0.70 | 0.22 | 0.43 | 0.36 | 0.70 | 0.22 | 0.73 | 0.26 | 0.78 | 0.22 | 0.73 | 0.22 |
| | | OCSVM | 0.39 | 0.07 | 0.42 | 0.04 | 0.73 | 0.06 | 0.43 | 0.04 | 0.40 | 0.07 | 0.41 | 0.07 | 0.40 | 0.05 |
| | | PEDM | 0.88 | 0.17 | 0.84 | 0.17 | 0.25 | 0.32 | 0.84 | 0.17 | 0.88 | 0.17 | 0.90 | 0.14 | 0.88 | 0.15 |
| | | RIQN | 0.25 | 0.03 | 0.37 | 0.01 | 0.84 | 0.03 | 0.37 | 0.01 | 0.27 | 0.03 | 0.30 | 0.04 | 0.35 | 0.02 |
| SAP-Goal0 | strong | IF | 0.54 | 0.27 | 0.73 | 0.16 | 0.64 | 0.33 | 0.80 | 0.11 | 0.70 | 0.18 | 0.61 | 0.26 | 0.78 | 0.17 |
| | | KNN | 0.83 | 0.21 | 0.86 | 0.16 | 0.31 | 0.31 | 0.89 | 0.12 | 0.88 | 0.13 | 0.90 | 0.13 | 0.93 | 0.11 |
| | | LSTM-DM | 0.93 | 0.16 | 0.95 | 0.08 | 0.11 | 0.24 | 0.96 | 0.06 | 0.95 | 0.12 | 0.95 | 0.15 | 0.97 | 0.06 |
| | | MLP-DM | 0.92 | 0.16 | 0.94 | 0.09 | 0.13 | 0.25 | 0.96 | 0.06 | 0.95 | 0.12 | 0.94 | 0.14 | 0.97 | 0.05 |
| | | OCSVM | 0.54 | 0.35 | 0.75 | 0.20 | 0.61 | 0.42 | 0.82 | 0.14 | 0.70 | 0.23 | 0.59 | 0.33 | 0.78 | 0.19 |
| | | PEDM | 0.93 | 0.16 | 0.96 | 0.07 | 0.12 | 0.24 | 0.97 | 0.06 | 0.95 | 0.11 | 0.95 | 0.14 | 0.98 | 0.05 |
| | | RIQN | 0.44 | 0.38 | 0.71 | 0.21 | 0.68 | 0.45 | 0.77 | 0.16 | 0.62 | 0.26 | 0.51 | 0.35 | 0.73 | 0.21 |
| | tiny | IF | 0.42 | 0.17 | 0.56 | 0.11 | 0.82 | 0.19 | 0.69 | 0.08 | 0.64 | 0.10 | 0.50 | 0.14 | 0.47 | 0.13 |
| | | KNN | 0.55 | 0.31 | 0.67 | 0.24 | 0.62 | 0.41 | 0.75 | 0.18 | 0.71 | 0.20 | 0.67 | 0.23 | 0.64 | 0.25 |
| | | LSTM-DM | 0.83 | 0.18 | 0.83 | 0.15 | 0.33 | 0.30 | 0.89 | 0.10 | 0.90 | 0.10 | 0.83 | 0.18 | 0.80 | 0.19 |

Table 10: Detailed results for all detectors and environments on vector observations. The mean and standard deviation are reported for each detector, environment, and anomaly strength. The results are grouped by environment and detector.

| env | strength | detector | AUROC mean | AUROC std | AUPR mean | AUPR std | FPR95 mean | FPR95 std | VUSPR mean | VUSPR std | VUSROC mean | VUSROC std | Global-AUROC mean | Global-AUROC std | Global-AUPR mean | Global-AUPR std |
|---|---|---|---|---|---|---|---|---|---|---|---|---|---|---|---|---|
| SAP-Goal1 | strong | MLP-DM | 0.79 | 0.19 | 0.81 | 0.16 | 0.41 | 0.33 | 0.87 | 0.11 | 0.88 | 0.11 | 0.79 | 0.19 | 0.77 | 0.20 |
| | | OCSVM | 0.50 | 0.32 | 0.64 | 0.23 | 0.66 | 0.41 | 0.75 | 0.16 | 0.69 | 0.19 | 0.56 | 0.28 | 0.56 | 0.28 |
| | | PEDM | 0.84 | 0.19 | 0.86 | 0.15 | 0.34 | 0.31 | 0.90 | 0.11 | 0.90 | 0.12 | 0.83 | 0.18 | 0.82 | 0.18 |
| | | RIQN | 0.42 | 0.36 | 0.60 | 0.25 | 0.71 | 0.44 | 0.70 | 0.19 | 0.63 | 0.23 | 0.51 | 0.31 | 0.53 | 0.29 |
| | | IF | 0.61 | 0.31 | 0.81 | 0.13 | 0.52 | 0.34 | 0.87 | 0.09 | 0.74 | 0.21 | 0.63 | 0.30 | 0.81 | 0.14 |
| | | KNN | 0.78 | 0.16 | 0.87 | 0.10 | 0.37 | 0.25 | 0.91 | 0.07 | 0.85 | 0.10 | 0.77 | 0.16 | 0.87 | 0.10 |
| | | LSTM-DM | 0.80 | 0.19 | 0.88 | 0.10 | 0.36 | 0.27 | 0.92 | 0.07 | 0.87 | 0.13 | 0.80 | 0.22 | 0.90 | 0.10 |
| | tiny | MLP-DM | 0.79 | 0.17 | 0.89 | 0.09 | 0.38 | 0.25 | 0.92 | 0.06 | 0.87 | 0.12 | 0.80 | 0.19 | 0.90 | 0.09 |
| | | OCSVM | 0.55 | 0.32 | 0.78 | 0.16 | 0.59 | 0.39 | 0.83 | 0.12 | 0.69 | 0.22 | 0.62 | 0.29 | 0.82 | 0.14 |
| | | PEDM | 0.84 | 0.19 | 0.91 | 0.08 | 0.29 | 0.26 | 0.94 | 0.06 | 0.89 | 0.13 | 0.84 | 0.22 | 0.92 | 0.09 |
| | | RIQN | 0.48 | 0.35 | 0.73 | 0.20 | 0.68 | 0.44 | 0.79 | 0.16 | 0.63 | 0.25 | 0.58 | 0.29 | 0.77 | 0.17 |
| | | IF | 0.64 | 0.18 | 0.75 | 0.13 | 0.54 | 0.22 | 0.82 | 0.08 | 0.77 | 0.11 | 0.66 | 0.16 | 0.67 | 0.16 |
| | | KNN | 0.75 | 0.16 | 0.81 | 0.13 | 0.43 | 0.27 | 0.87 | 0.09 | 0.84 | 0.10 | 0.73 | 0.18 | 0.71 | 0.19 |
| | | LSTM-DM | 0.80 | 0.15 | 0.83 | 0.13 | 0.38 | 0.27 | 0.89 | 0.08 | 0.88 | 0.09 | 0.77 | 0.16 | 0.76 | 0.17 |
| SAP-Goal2 | strong | MLP-DM | 0.79 | 0.14 | 0.83 | 0.12 | 0.43 | 0.26 | 0.89 | 0.08 | 0.87 | 0.09 | 0.76 | 0.15 | 0.75 | 0.16 |
| | | OCSVM | 0.57 | 0.29 | 0.71 | 0.19 | 0.57 | 0.36 | 0.78 | 0.14 | 0.70 | 0.19 | 0.64 | 0.24 | 0.67 | 0.22 |
| | | PEDM | 0.83 | 0.14 | 0.85 | 0.11 | 0.35 | 0.26 | 0.90 | 0.08 | 0.89 | 0.09 | 0.81 | 0.14 | 0.78 | 0.16 |
| | | RIQN | 0.44 | 0.35 | 0.63 | 0.24 | 0.69 | 0.43 | 0.70 | 0.19 | 0.62 | 0.24 | 0.54 | 0.29 | 0.58 | 0.27 |
| | | IF | 0.37 | 0.33 | 0.66 | 0.17 | 0.76 | 0.32 | 0.72 | 0.13 | 0.53 | 0.24 | 0.49 | 0.31 | 0.69 | 0.19 |
| | | KNN | 0.68 | 0.27 | 0.80 | 0.18 | 0.47 | 0.38 | 0.83 | 0.14 | 0.77 | 0.19 | 0.74 | 0.21 | 0.83 | 0.15 |
| | | LSTM-DM | 0.78 | 0.19 | 0.84 | 0.13 | 0.38 | 0.29 | 0.88 | 0.09 | 0.85 | 0.13 | 0.77 | 0.23 | 0.85 | 0.14 |
| | tiny | MLP-DM | 0.74 | 0.20 | 0.82 | 0.12 | 0.45 | 0.30 | 0.88 | 0.09 | 0.83 | 0.14 | 0.71 | 0.24 | 0.82 | 0.14 |
| | | OCSVM | 0.45 | 0.37 | 0.70 | 0.21 | 0.66 | 0.43 | 0.76 | 0.17 | 0.59 | 0.28 | 0.54 | 0.32 | 0.73 | 0.19 |
| | | PEDM | 0.79 | 0.21 | 0.86 | 0.12 | 0.38 | 0.32 | 0.90 | 0.09 | 0.86 | 0.15 | 0.79 | 0.25 | 0.88 | 0.13 |
| | | RIQN | 0.52 | 0.38 | 0.74 | 0.22 | 0.60 | 0.43 | 0.78 | 0.18 | 0.64 | 0.28 | 0.62 | 0.30 | 0.76 | 0.20 |
| | | IF | 0.25 | 0.14 | 0.49 | 0.08 | 0.90 | 0.08 | 0.58 | 0.06 | 0.46 | 0.09 | 0.42 | 0.12 | 0.42 | 0.09 |
| | | KNN | 0.55 | 0.30 | 0.67 | 0.22 | 0.60 | 0.40 | 0.74 | 0.18 | 0.69 | 0.21 | 0.62 | 0.25 | 0.62 | 0.26 |
| | | LSTM-DM | 0.78 | 0.16 | 0.78 | 0.16 | 0.40 | 0.29 | 0.84 | 0.12 | 0.86 | 0.10 | 0.75 | 0.18 | 0.70 | 0.23 |
| URM-PnP | strong | MLP-DM | 0.77 | 0.16 | 0.77 | 0.15 | 0.45 | 0.29 | 0.84 | 0.11 | 0.86 | 0.10 | 0.73 | 0.18 | 0.67 | 0.22 |
| | | OCSVM | 0.44 | 0.38 | 0.61 | 0.26 | 0.65 | 0.43 | 0.68 | 0.21 | 0.60 | 0.27 | 0.55 | 0.31 | 0.56 | 0.29 |
| | | PEDM | 0.78 | 0.17 | 0.79 | 0.16 | 0.42 | 0.32 | 0.85 | 0.11 | 0.87 | 0.11 | 0.75 | 0.19 | 0.70 | 0.23 |
| | | RIQN | 0.42 | 0.38 | 0.60 | 0.26 | 0.66 | 0.43 | 0.67 | 0.22 | 0.58 | 0.27 | 0.55 | 0.30 | 0.56 | 0.29 |
| | | IF | 0.42 | 0.25 | 0.53 | 0.15 | 0.79 | 0.25 | 0.53 | 0.14 | 0.43 | 0.24 | 0.54 | 0.19 | 0.57 | 0.16 |
| | | KNN | 0.68 | 0.31 | 0.72 | 0.25 | 0.47 | 0.40 | 0.73 | 0.25 | 0.68 | 0.31 | 0.72 | 0.27 | 0.76 | 0.24 |
| | | LSTM-DM | 0.77 | 0.30 | 0.76 | 0.22 | 0.32 | 0.38 | 0.76 | 0.22 | 0.77 | 0.30 | 0.83 | 0.23 | 0.82 | 0.18 |
| | tiny | MLP-DM | 0.77 | 0.31 | 0.78 | 0.23 | 0.32 | 0.39 | 0.78 | 0.23 | 0.78 | 0.30 | 0.83 | 0.24 | 0.84 | 0.19 |
| | | OCSVM | 0.57 | 0.37 | 0.66 | 0.29 | 0.52 | 0.44 | 0.67 | 0.28 | 0.58 | 0.36 | 0.65 | 0.31 | 0.68 | 0.28 |
| | | PEDM | 0.83 | 0.29 | 0.84 | 0.22 | 0.26 | 0.38 | 0.84 | 0.22 | 0.83 | 0.28 | 0.87 | 0.22 | 0.89 | 0.17 |
| | | RIQN | 0.57 | 0.37 | 0.66 | 0.29 | 0.56 | 0.47 | 0.67 | 0.28 | 0.58 | 0.36 | 0.64 | 0.31 | 0.68 | 0.28 |
| | | IF | 0.24 | 0.16 | 0.44 | 0.07 | 0.89 | 0.15 | 0.44 | 0.07 | 0.26 | 0.16 | 0.37 | 0.12 | 0.43 | 0.08 |
| | | KNN | 0.58 | 0.37 | 0.67 | 0.28 | 0.51 | 0.43 | 0.67 | 0.28 | 0.59 | 0.36 | 0.62 | 0.33 | 0.66 | 0.29 |
| | | LSTM-DM | 0.63 | 0.36 | 0.68 | 0.26 | 0.46 | 0.42 | 0.68 | 0.25 | 0.64 | 0.35 | 0.71 | 0.31 | 0.71 | 0.26 |
| URM-Reach | strong | MLP-DM | 0.63 | 0.37 | 0.69 | 0.27 | 0.47 | 0.43 | 0.70 | 0.27 | 0.64 | 0.36 | 0.71 | 0.31 | 0.72 | 0.27 |
| | | OCSVM | 0.51 | 0.41 | 0.64 | 0.30 | 0.53 | 0.45 | 0.64 | 0.30 | 0.53 | 0.40 | 0.60 | 0.34 | 0.64 | 0.31 |
| | | PEDM | 0.76 | 0.34 | 0.78 | 0.24 | 0.35 | 0.40 | 0.78 | 0.24 | 0.76 | 0.33 | 0.80 | 0.29 | 0.81 | 0.24 |
| | | RIQN | 0.53 | 0.40 | 0.64 | 0.30 | 0.57 | 0.48 | 0.65 | 0.30 | 0.54 | 0.39 | 0.60 | 0.34 | 0.64 | 0.30 |
| | | IF | 0.57 | 0.10 | 0.59 | 0.06 | 0.74 | 0.10 | 0.60 | 0.06 | 0.59 | 0.10 | 0.59 | 0.10 | 0.58 | 0.09 |
| | | KNN | 0.82 | 0.17 | 0.80 | 0.18 | 0.39 | 0.34 | 0.80 | 0.18 | 0.82 | 0.16 | 0.83 | 0.16 | 0.82 | 0.17 |
| | | LSTM-DM | 0.93 | 0.09 | 0.87 | 0.12 | 0.20 | 0.26 | 0.87 | 0.11 | 0.94 | 0.08 | 0.94 | 0.08 | 0.91 | 0.09 |
| | tiny | MLP-DM | 0.93 | 0.10 | 0.88 | 0.13 | 0.19 | 0.23 | 0.88 | 0.12 | 0.93 | 0.10 | 0.93 | 0.10 | 0.91 | 0.11 |
| | | OCSVM | 0.70 | 0.26 | 0.73 | 0.23 | 0.47 | 0.40 | 0.73 | 0.23 | 0.71 | 0.25 | 0.71 | 0.26 | 0.71 | 0.25 |
| | | PEDM | 0.97 | 0.04 | 0.92 | 0.08 | 0.11 | 0.14 | 0.93 | 0.08 | 0.97 | 0.04 | 0.97 | 0.04 | 0.96 | 0.05 |
| | | RIQN | 0.69 | 0.27 | 0.73 | 0.23 | 0.48 | 0.41 | 0.73 | 0.23 | 0.71 | 0.25 | 0.70 | 0.26 | 0.71 | 0.25 |
| | | IF | 0.51 | 0.06 | 0.55 | 0.04 | 0.79 | 0.07 | 0.56 | 0.04 | 0.53 | 0.06 | 0.52 | 0.05 | 0.52 | 0.05 |
| | | KNN | 0.74 | 0.22 | 0.75 | 0.21 | 0.46 | 0.39 | 0.76 | 0.21 | 0.75 | 0.21 | 0.74 | 0.22 | 0.75 | 0.22 |
| | | LSTM-DM | 0.87 | 0.17 | 0.83 | 0.16 | 0.29 | 0.36 | 0.83 | 0.16 | 0.88 | 0.16 | 0.88 | 0.16 | 0.85 | 0.16 |
| URR-Reach | strong | MLP-DM | 0.86 | 0.17 | 0.83 | 0.17 | 0.30 | 0.35 | 0.83 | 0.16 | 0.87 | 0.16 | 0.86 | 0.18 | 0.84 | 0.18 |
| | | OCSVM | 0.68 | 0.27 | 0.72 | 0.24 | 0.48 | 0.41 | 0.72 | 0.23 | 0.70 | 0.26 | 0.69 | 0.26 | 0.70 | 0.26 |
| | | PEDM | 0.91 | 0.13 | 0.87 | 0.14 | 0.23 | 0.30 | 0.87 | 0.14 | 0.91 | 0.13 | 0.91 | 0.14 | 0.89 | 0.14 |
| | | RIQN | 0.68 | 0.27 | 0.72 | 0.24 | 0.49 | 0.41 | 0.73 | 0.23 | 0.69 | 0.26 | 0.69 | 0.26 | 0.69 | 0.26 |
| | | IF | 0.57 | 0.13 | 0.57 | 0.09 | 0.77 | 0.09 | 0.58 | 0.09 | 0.58 | 0.12 | 0.57 | 0.13 | 0.57 | 0.15 |
| | | KNN | 0.77 | 0.19 | 0.75 | 0.21 | 0.46 | 0.37 | 0.75 | 0.20 | 0.78 | 0.18 | 0.79 | 0.18 | 0.77 | 0.20 |
| | | LSTM-DM | 0.90 | 0.12 | 0.82 | 0.13 | 0.26 | 0.30 | 0.83 | 0.13 | 0.91 | 0.12 | 0.90 | 0.13 | 0.86 | 0.14 |
| | tiny | MLP-DM | 0.89 | 0.15 | 0.83 | 0.17 | 0.29 | 0.33 | 0.83 | 0.16 | 0.89 | 0.14 | 0.89 | 0.15 | 0.86 | 0.16 |
| | | OCSVM | 0.68 | 0.26 | 0.70 | 0.25 | 0.51 | 0.41 | 0.70 | 0.24 | 0.69 | 0.26 | 0.68 | 0.27 | 0.68 | 0.27 |
| | | PEDM | 0.95 | 0.07 | 0.88 | 0.13 | 0.15 | 0.17 | 0.88 | 0.12 | 0.95 | 0.07 | 0.94 | 0.09 | 0.91 | 0.12 |
| | | RIQN | 0.68 | 0.26 | 0.70 | 0.25 | 0.52 | 0.41 | 0.70 | 0.24 | 0.69 | 0.25 | 0.67 | 0.28 | 0.66 | 0.28 |
| | | IF | 0.49 | 0.06 | 0.52 | 0.04 | 0.82 | 0.04 | 0.53 | 0.04 | 0.50 | 0.06 | 0.49 | 0.05 | 0.48 | 0.05 |
| | | KNN | 0.70 | 0.26 | 0.72 | 0.24 | 0.50 | 0.43 | 0.72 | 0.24 | 0.71 | 0.25 | 0.71 | 0.25 | 0.70 | 0.26 |
| | | LSTM-DM | 0.84 | 0.19 | 0.79 | 0.17 | 0.36 | 0.39 | 0.79 | 0.17 | 0.85 | 0.18 | 0.83 | 0.20 | 0.81 | 0.19 |
| | | MLP-DM | 0.82 | 0.21 | 0.79 | 0.21 | 0.39 | 0.42 | 0.79 | 0.20 | 0.83 | 0.20 | 0.82 | 0.22 | 0.80 | 0.22 |
| | | OCSVM | 0.68 | 0.27 | 0.71 | 0.25 | 0.49 | 0.42 | 0.71 | 0.25 | 0.70 | 0.26 | 0.67 | 0.28 | 0.67 | 0.28 |
| | | PEDM | 0.86 | 0.19 | 0.83 | 0.19 | 0.30 | 0.38 | 0.83 | 0.19 | 0.87 | 0.18 | 0.85 | 0.21 | 0.84 | 0.21 |
| | | RIQN | 0.69 | 0.27 | 0.71 | 0.25 | 0.50 | 0.43 | 0.71 | 0.25 | 0.70 | 0.26 | 0.67 | 0.28 | 0.67 | 0.28 |

Table 11: Detailed results for all detectors and environments on img observations. The mean and standard deviation are reported for each detector, environment, and anomaly strength. The results are grouped by environment and detector.

| env | strength | detector | AUROC mean | std | AUPR mean | std | FPR95 mean | std | VUSPR mean | std | VUSROC mean | std | Global-AUROC mean | std | Global-AUPR mean | std |
|---|---|---|---|---|---|---|---|---|---|---|---|---|---|---|---|---|
| CAR-LaneKeep | strong | AE | 0.51 | 0.06 | 0.54 | 0.05 | 0.81 | 0.04 | 0.54 | 0.05 | 0.53 | 0.06 | 0.55 | 0.06 | 0.51 | 0.07 |
| | | AE-KNN | 0.54 | 0.07 | 0.54 | 0.05 | 0.75 | 0.05 | 0.55 | 0.05 | 0.55 | 0.07 | 0.57 | 0.07 | 0.49 | 0.06 |
| | | CLIP-KNN | 0.53 | 0.07 | 0.52 | 0.05 | 0.84 | 0.05 | 0.53 | 0.05 | 0.54 | 0.07 | 0.54 | 0.07 | 0.48 | 0.06 |
| | | Dino-Patch | 0.52 | 0.02 | 0.52 | 0.03 | 0.88 | 0.02 | 0.52 | 0.02 | 0.54 | 0.02 | 0.52 | 0.02 | 0.46 | 0.03 |
| | | L-DM | 0.57 | 0.13 | 0.58 | 0.09 | 0.75 | 0.10 | 0.59 | 0.09 | 0.59 | 0.13 | 0.60 | 0.12 | 0.55 | 0.13 |
| | | L-RDM | 0.58 | 0.14 | 0.59 | 0.10 | 0.74 | 0.12 | 0.59 | 0.10 | 0.59 | 0.13 | 0.61 | 0.13 | 0.55 | 0.13 |
| | | Pred-AE | 0.68 | 0.15 | 0.64 | 0.13 | 0.68 | 0.14 | 0.64 | 0.13 | 0.69 | 0.14 | 0.66 | 0.16 | 0.61 | 0.18 |
| | | PredNet | 0.58 | 0.08 | 0.55 | 0.06 | 0.76 | 0.07 | 0.56 | 0.06 | 0.60 | 0.08 | 0.58 | 0.08 | 0.51 | 0.08 |
| | | RES-KNN | 0.46 | 0.07 | 0.48 | 0.05 | 0.83 | 0.05 | 0.49 | 0.05 | 0.48 | 0.07 | 0.51 | 0.06 | 0.43 | 0.05 |
| | tiny | AE | 0.43 | 0.02 | 0.51 | 0.01 | 0.85 | 0.02 | 0.51 | 0.01 | 0.45 | 0.02 | 0.47 | 0.02 | 0.46 | 0.02 |
| | | AE-KNN | 0.44 | 0.02 | 0.51 | 0.01 | 0.81 | 0.02 | 0.51 | 0.01 | 0.46 | 0.02 | 0.47 | 0.01 | 0.45 | 0.01 |
| | | CLIP-KNN | 0.41 | 0.03 | 0.48 | 0.01 | 0.90 | 0.01 | 0.48 | 0.01 | 0.42 | 0.03 | 0.43 | 0.02 | 0.43 | 0.01 |
| | | Dino-Patch | 0.50 | 0.01 | 0.53 | 0.01 | 0.88 | 0.01 | 0.53 | 0.01 | 0.51 | 0.01 | 0.51 | 0.01 | 0.48 | 0.02 |
| | | L-DM | 0.41 | 0.03 | 0.50 | 0.02 | 0.86 | 0.02 | 0.50 | 0.02 | 0.42 | 0.03 | 0.45 | 0.02 | 0.44 | 0.02 |
| | | L-RDM | 0.40 | 0.04 | 0.49 | 0.02 | 0.86 | 0.02 | 0.50 | 0.02 | 0.42 | 0.03 | 0.45 | 0.03 | 0.44 | 0.02 |
| | | Pred-AE | 0.55 | 0.04 | 0.54 | 0.02 | 0.76 | 0.04 | 0.54 | 0.02 | 0.56 | 0.04 | 0.51 | 0.04 | 0.48 | 0.04 |
| | | PredNet | 0.52 | 0.02 | 0.53 | 0.01 | 0.82 | 0.01 | 0.54 | 0.01 | 0.53 | 0.02 | 0.52 | 0.01 | 0.49 | 0.02 |
| | | RES-KNN | 0.36 | 0.02 | 0.46 | 0.02 | 0.89 | 0.02 | 0.46 | 0.02 | 0.38 | 0.02 | 0.42 | 0.02 | 0.41 | 0.01 |
| MJC-CartpoleSwingup | strong | AE | 0.73 | 0.13 | 0.71 | 0.12 | 0.72 | 0.20 | 0.72 | 0.12 | 0.75 | 0.12 | 0.74 | 0.16 | 0.79 | 0.14 |
| | | AE-KNN | 0.76 | 0.13 | 0.76 | 0.12 | 0.73 | 0.20 | 0.76 | 0.12 | 0.78 | 0.13 | 0.80 | 0.13 | 0.85 | 0.11 |
| | | CLIP-KNN | 0.75 | 0.12 | 0.73 | 0.11 | 0.71 | 0.20 | 0.74 | 0.11 | 0.77 | 0.11 | 0.73 | 0.14 | 0.78 | 0.13 |
| | | Dino-Patch | 0.63 | 0.08 | 0.57 | 0.06 | 0.66 | 0.09 | 0.58 | 0.06 | 0.65 | 0.08 | 0.50 | 0.11 | 0.52 | 0.09 |
| | | L-DM | 0.84 | 0.14 | 0.82 | 0.14 | 0.50 | 0.31 | 0.82 | 0.14 | 0.85 | 0.13 | 0.86 | 0.14 | 0.90 | 0.11 |
| | | L-RDM | 0.87 | 0.10 | 0.83 | 0.13 | 0.42 | 0.25 | 0.83 | 0.13 | 0.87 | 0.10 | 0.88 | 0.11 | 0.91 | 0.09 |
| | | Pred-AE | 0.81 | 0.15 | 0.80 | 0.14 | 0.53 | 0.31 | 0.80 | 0.14 | 0.82 | 0.15 | 0.84 | 0.14 | 0.87 | 0.12 |
| | | PredNet | 0.68 | 0.14 | 0.63 | 0.12 | 0.60 | 0.13 | 0.63 | 0.12 | 0.69 | 0.14 | 0.72 | 0.14 | 0.68 | 0.14 |
| | | RES-KNN | 0.79 | 0.10 | 0.75 | 0.12 | 0.64 | 0.21 | 0.75 | 0.12 | 0.80 | 0.10 | 0.78 | 0.13 | 0.82 | 0.12 |
| | tiny | AE | 0.52 | 0.17 | 0.54 | 0.08 | 0.83 | 0.16 | 0.55 | 0.08 | 0.54 | 0.16 | 0.43 | 0.16 | 0.50 | 0.08 |
| | | AE-KNN | 0.55 | 0.17 | 0.58 | 0.10 | 0.85 | 0.13 | 0.59 | 0.10 | 0.57 | 0.16 | 0.55 | 0.19 | 0.61 | 0.14 |
| | | CLIP-KNN | 0.57 | 0.19 | 0.57 | 0.11 | 0.78 | 0.19 | 0.58 | 0.11 | 0.59 | 0.18 | 0.50 | 0.20 | 0.55 | 0.12 |
| | | Dino-Patch | 0.72 | 0.08 | 0.63 | 0.09 | 0.52 | 0.08 | 0.64 | 0.09 | 0.74 | 0.08 | 0.64 | 0.13 | 0.62 | 0.14 |
| | | L-DM | 0.59 | 0.21 | 0.62 | 0.15 | 0.78 | 0.22 | 0.62 | 0.14 | 0.61 | 0.21 | 0.57 | 0.24 | 0.64 | 0.17 |
| | | L-RDM | 0.69 | 0.20 | 0.67 | 0.16 | 0.65 | 0.25 | 0.68 | 0.15 | 0.71 | 0.20 | 0.65 | 0.24 | 0.72 | 0.18 |
| | | Pred-AE | 0.48 | 0.24 | 0.56 | 0.14 | 0.81 | 0.23 | 0.57 | 0.14 | 0.51 | 0.23 | 0.47 | 0.25 | 0.55 | 0.18 |
| | | PredNet | 0.52 | 0.11 | 0.49 | 0.05 | 0.72 | 0.07 | 0.50 | 0.05 | 0.54 | 0.11 | 0.56 | 0.15 | 0.51 | 0.08 |
| | | RES-KNN | 0.61 | 0.16 | 0.57 | 0.09 | 0.73 | 0.19 | 0.58 | 0.09 | 0.63 | 0.15 | 0.54 | 0.18 | 0.57 | 0.12 |
| MJC-HalfCheetah | strong | AE | 0.42 | 0.02 | 0.51 | 0.03 | 0.98 | 0.01 | 0.52 | 0.03 | 0.45 | 0.02 | 0.43 | 0.02 | 0.49 | 0.03 |
| | | AE-KNN | 0.51 | 0.01 | 0.56 | 0.02 | 0.95 | 0.01 | 0.57 | 0.02 | 0.53 | 0.01 | 0.49 | 0.01 | 0.56 | 0.02 |
| | | CLIP-KNN | 0.61 | 0.03 | 0.61 | 0.03 | 0.86 | 0.03 | 0.62 | 0.03 | 0.63 | 0.03 | 0.58 | 0.03 | 0.61 | 0.03 |
| | | Dino-Patch | 0.53 | 0.02 | 0.57 | 0.03 | 0.94 | 0.01 | 0.58 | 0.03 | 0.55 | 0.01 | 0.53 | 0.02 | 0.57 | 0.03 |
| | | L-DM | 0.23 | 0.04 | 0.42 | 0.02 | 0.95 | 0.03 | 0.43 | 0.02 | 0.27 | 0.04 | 0.26 | 0.03 | 0.39 | 0.02 |
| | | L-RDM | 0.63 | 0.04 | 0.62 | 0.04 | 0.81 | 0.03 | 0.62 | 0.04 | 0.65 | 0.04 | 0.65 | 0.04 | 0.68 | 0.05 |
| | | Pred-AE | 0.86 | 0.04 | 0.74 | 0.06 | 0.30 | 0.07 | 0.76 | 0.06 | 0.88 | 0.04 | 0.63 | 0.05 | 0.58 | 0.06 |
| | | PredNet | 0.58 | 0.05 | 0.59 | 0.04 | 0.83 | 0.02 | 0.60 | 0.04 | 0.61 | 0.05 | 0.58 | 0.05 | 0.60 | 0.05 |
| | | RES-KNN | 0.66 | 0.02 | 0.66 | 0.03 | 0.86 | 0.01 | 0.67 | 0.03 | 0.69 | 0.02 | 0.63 | 0.02 | 0.66 | 0.04 |
| | tiny | AE | 0.47 | 0.01 | 0.55 | 0.01 | 0.95 | 0.01 | 0.55 | 0.01 | 0.50 | 0.01 | 0.48 | 0.01 | 0.53 | 0.01 |
| | | AE-KNN | 0.49 | 0.01 | 0.56 | 0.01 | 0.94 | 0.01 | 0.57 | 0.01 | 0.52 | 0.01 | 0.48 | 0.01 | 0.55 | 0.01 |
| | | CLIP-KNN | 0.57 | 0.01 | 0.57 | 0.01 | 0.86 | 0.02 | 0.58 | 0.01 | 0.60 | 0.01 | 0.55 | 0.01 | 0.56 | 0.01 |
| | | Dino-Patch | 0.50 | 0.01 | 0.56 | 0.01 | 0.94 | 0.00 | 0.57 | 0.01 | 0.53 | 0.01 | 0.50 | 0.01 | 0.55 | 0.01 |
| | | L-DM | 0.34 | 0.01 | 0.46 | 0.01 | 0.84 | 0.01 | 0.47 | 0.01 | 0.38 | 0.01 | 0.41 | 0.01 | 0.44 | 0.01 |
| | | L-RDM | 0.56 | 0.00 | 0.58 | 0.01 | 0.87 | 0.01 | 0.59 | 0.01 | 0.58 | 0.01 | 0.53 | 0.01 | 0.56 | 0.01 |
| | | Pred-AE | 0.96 | 0.01 | 0.87 | 0.02 | 0.12 | 0.02 | 0.88 | 0.02 | 0.97 | 0.00 | 0.82 | 0.02 | 0.80 | 0.02 |
| | | PredNet | 0.46 | 0.03 | 0.50 | 0.02 | 0.84 | 0.02 | 0.51 | 0.02 | 0.49 | 0.03 | 0.46 | 0.04 | 0.49 | 0.03 |
| | | RES-KNN | 0.61 | 0.01 | 0.62 | 0.01 | 0.88 | 0.01 | 0.63 | 0.01 | 0.64 | 0.01 | 0.58 | 0.01 | 0.60 | 0.01 |
| MJC-Reacher3D | strong | AE | 0.38 | 0.07 | 0.43 | 0.04 | 0.75 | 0.02 | 0.44 | 0.04 | 0.40 | 0.07 | 0.44 | 0.10 | 0.44 | 0.11 |
| | | AE-KNN | 0.47 | 0.03 | 0.46 | 0.02 | 0.82 | 0.02 | 0.47 | 0.02 | 0.49 | 0.03 | 0.50 | 0.05 | 0.47 | 0.05 |
| | | CLIP-KNN | 0.84 | 0.06 | 0.74 | 0.07 | 0.41 | 0.06 | 0.75 | 0.07 | 0.85 | 0.06 | 0.78 | 0.10 | 0.77 | 0.12 |
| | | Dino-Patch | 0.53 | 0.08 | 0.53 | 0.06 | 0.74 | 0.03 | 0.53 | 0.06 | 0.55 | 0.08 | 0.58 | 0.08 | 0.58 | 0.09 |
| | | L-DM | 0.86 | 0.05 | 0.76 | 0.08 | 0.31 | 0.03 | 0.76 | 0.08 | 0.86 | 0.05 | 0.81 | 0.08 | 0.79 | 0.11 |
| | | L-RDM | 0.92 | 0.02 | 0.84 | 0.04 | 0.16 | 0.02 | 0.85 | 0.03 | 0.93 | 0.02 | 0.84 | 0.05 | 0.82 | 0.07 |
| | | Pred-AE | 0.49 | 0.13 | 0.51 | 0.09 | 0.83 | 0.05 | 0.51 | 0.09 | 0.51 | 0.13 | 0.53 | 0.12 | 0.54 | 0.11 |
| | | PredNet | 0.63 | 0.11 | 0.59 | 0.09 | 0.73 | 0.06 | 0.60 | 0.09 | 0.64 | 0.11 | 0.63 | 0.12 | 0.62 | 0.12 |
| | | RES-KNN | 0.82 | 0.06 | 0.71 | 0.07 | 0.47 | 0.07 | 0.72 | 0.07 | 0.83 | 0.06 | 0.79 | 0.09 | 0.76 | 0.11 |
| | tiny | AE | 0.34 | 0.01 | 0.41 | 0.01 | 0.74 | 0.01 | 0.41 | 0.01 | 0.35 | 0.01 | 0.38 | 0.02 | 0.38 | 0.01 |
| | | AE-KNN | 0.45 | 0.01 | 0.45 | 0.01 | 0.81 | 0.01 | 0.46 | 0.01 | 0.48 | 0.01 | 0.47 | 0.01 | 0.44 | 0.01 |
| | | CLIP-KNN | 0.75 | 0.02 | 0.66 | 0.02 | 0.50 | 0.02 | 0.67 | 0.02 | 0.77 | 0.02 | 0.64 | 0.03 | 0.57 | 0.04 |
| | | Dino-Patch | 0.43 | 0.02 | 0.47 | 0.01 | 0.78 | 0.02 | 0.48 | 0.01 | 0.45 | 0.02 | 0.46 | 0.02 | 0.46 | 0.02 |
| | | L-DM | 0.79 | 0.02 | 0.67 | 0.02 | 0.37 | 0.02 | 0.68 | 0.02 | 0.80 | 0.02 | 0.68 | 0.02 | 0.60 | 0.04 |
| | | L-RDM | 0.91 | 0.01 | 0.82 | 0.01 | 0.17 | 0.01 | 0.82 | 0.01 | 0.91 | 0.01 | 0.75 | 0.02 | 0.67 | 0.05 |
| | | Pred-AE | 0.33 | 0.02 | 0.41 | 0.01 | 0.90 | 0.01 | 0.42 | 0.02 | 0.36 | 0.02 | 0.36 | 0.02 | 0.39 | 0.01 |
| | | PredNet | 0.53 | 0.02 | 0.49 | 0.02 | 0.76 | 0.01 | 0.50 | 0.01 | 0.55 | 0.02 | 0.54 | 0.03 | 0.48 | 0.02 |
| | | RES-KNN | 0.73 | 0.02 | 0.62 | 0.02 | 0.57 | 0.02 | 0.64 | 0.02 | 0.75 | 0.02 | 0.63 | 0.02 | 0.56 | 0.04 |
| SAP-Goal0 | strong | AE | 0.73 | 0.06 | 0.81 | 0.06 | 0.71 | 0.07 | 0.89 | 0.03 | 0.85 | 0.03 | 0.56 | 0.03 | 0.73 | 0.11 |
| | | AE-KNN | 0.58 | 0.03 | 0.74 | 0.05 | 0.70 | 0.04 | 0.82 | 0.03 | 0.73 | 0.03 | 0.53 | 0.02 | 0.73 | 0.09 |
| | | CLIP-KNN | 0.71 | 0.03 | 0.82 | 0.05 | 0.64 | 0.03 | 0.89 | 0.03 | 0.83 | 0.01 | 0.64 | 0.04 | 0.79 | 0.10 |

Table 11: Detailed results for all detectors and environments on img observations. The mean and standard deviation are reported for each detector, environment, and anomaly strength. The results are grouped by environment and detector.

| env | strength | detector | AUROC mean | std | AUPR mean | std | FPR95 mean | std | VUSPR mean | std | VUSROC mean | std | Global-AUROC mean | std | Global-AUPR mean | std |
|---|---|---|---|---|---|---|---|---|---|---|---|---|---|---|---|---|
| | | Dino-Patch | 0.63 | 0.04 | 0.79 | 0.06 | 0.74 | 0.04 | 0.87 | 0.04 | 0.78 | 0.02 | 0.61 | 0.04 | 0.78 | 0.10 |
| | | L-DM | 0.62 | 0.10 | 0.75 | 0.09 | 0.82 | 0.11 | 0.83 | 0.05 | 0.78 | 0.05 | 0.60 | 0.04 | 0.75 | 0.11 |
| | | L-RDM | 0.67 | 0.08 | 0.76 | 0.08 | 0.77 | 0.09 | 0.85 | 0.05 | 0.83 | 0.04 | 0.64 | 0.05 | 0.81 | 0.11 |
| | | Pred-AE | 0.62 | 0.08 | 0.74 | 0.08 | 0.77 | 0.07 | 0.80 | 0.06 | 0.76 | 0.05 | 0.72 | 0.10 | 0.86 | 0.10 |
| | | PredNet | 0.49 | 0.07 | 0.65 | 0.09 | 0.85 | 0.05 | 0.71 | 0.07 | 0.65 | 0.04 | 0.60 | 0.07 | 0.72 | 0.12 |
| | | RES-KNN | 0.68 | 0.09 | 0.77 | 0.08 | 0.65 | 0.10 | 0.83 | 0.05 | 0.80 | 0.05 | 0.75 | 0.09 | 0.86 | 0.10 |
| | tiny | AE | 0.64 | 0.02 | 0.72 | 0.02 | 0.75 | 0.03 | 0.83 | 0.02 | 0.81 | 0.02 | 0.51 | 0.02 | 0.47 | 0.02 |
| | | AE-KNN | 0.61 | 0.03 | 0.67 | 0.01 | 0.63 | 0.03 | 0.78 | 0.01 | 0.77 | 0.02 | 0.51 | 0.01 | 0.49 | 0.03 |
| | | CLIP-KNN | 0.67 | 0.01 | 0.74 | 0.01 | 0.65 | 0.02 | 0.85 | 0.01 | 0.82 | 0.01 | 0.58 | 0.01 | 0.54 | 0.03 |
| | | Dino-Patch | 0.56 | 0.02 | 0.67 | 0.02 | 0.78 | 0.03 | 0.79 | 0.01 | 0.74 | 0.01 | 0.54 | 0.01 | 0.52 | 0.03 |
| | | L-DM | 0.45 | 0.03 | 0.61 | 0.02 | 0.95 | 0.02 | 0.75 | 0.01 | 0.70 | 0.02 | 0.51 | 0.01 | 0.47 | 0.02 |
| | | L-RDM | 0.54 | 0.02 | 0.65 | 0.02 | 0.93 | 0.02 | 0.79 | 0.01 | 0.77 | 0.01 | 0.53 | 0.01 | 0.50 | 0.03 |
| | | Pred-AE | 0.48 | 0.03 | 0.59 | 0.02 | 0.81 | 0.02 | 0.70 | 0.02 | 0.68 | 0.02 | 0.52 | 0.04 | 0.50 | 0.05 |
| | | PredNet | 0.38 | 0.05 | 0.50 | 0.02 | 0.86 | 0.07 | 0.61 | 0.01 | 0.60 | 0.03 | 0.43 | 0.04 | 0.41 | 0.03 |
| | | RES-KNN | 0.52 | 0.02 | 0.62 | 0.02 | 0.76 | 0.03 | 0.74 | 0.01 | 0.71 | 0.02 | 0.56 | 0.03 | 0.52 | 0.04 |
| SAP-Goal1 | strong | AE | 0.73 | 0.06 | 0.80 | 0.08 | 0.71 | 0.04 | 0.87 | 0.05 | 0.85 | 0.03 | 0.59 | 0.04 | 0.82 | 0.09 |
| | | AE-KNN | 0.50 | 0.02 | 0.78 | 0.05 | 0.74 | 0.04 | 0.83 | 0.03 | 0.66 | 0.02 | 0.52 | 0.02 | 0.74 | 0.07 |
| | | CLIP-KNN | 0.66 | 0.02 | 0.83 | 0.04 | 0.72 | 0.04 | 0.88 | 0.03 | 0.78 | 0.02 | 0.64 | 0.03 | 0.82 | 0.04 |
| | | Dino-Patch | 0.58 | 0.02 | 0.81 | 0.05 | 0.78 | 0.03 | 0.86 | 0.03 | 0.73 | 0.02 | 0.56 | 0.03 | 0.77 | 0.06 |
| | | L-DM | 0.57 | 0.07 | 0.75 | 0.08 | 0.80 | 0.09 | 0.82 | 0.06 | 0.73 | 0.04 | 0.51 | 0.02 | 0.74 | 0.08 |
| | | L-RDM | 0.70 | 0.09 | 0.80 | 0.08 | 0.78 | 0.09 | 0.88 | 0.05 | 0.85 | 0.04 | 0.57 | 0.03 | 0.78 | 0.09 |
| | | Pred-AE | 0.39 | 0.15 | 0.69 | 0.08 | 0.96 | 0.04 | 0.78 | 0.06 | 0.62 | 0.11 | 0.42 | 0.20 | 0.75 | 0.11 |
| | | PredNet | 0.51 | 0.05 | 0.70 | 0.09 | 0.87 | 0.01 | 0.75 | 0.07 | 0.67 | 0.02 | 0.59 | 0.03 | 0.74 | 0.08 |
| | | RES-KNN | 0.53 | 0.02 | 0.74 | 0.06 | 0.78 | 0.02 | 0.81 | 0.04 | 0.68 | 0.01 | 0.52 | 0.02 | 0.74 | 0.07 |
| | tiny | AE | 0.63 | 0.04 | 0.68 | 0.02 | 0.78 | 0.02 | 0.79 | 0.02 | 0.79 | 0.03 | 0.52 | 0.04 | 0.58 | 0.07 |
| | | AE-KNN | 0.52 | 0.02 | 0.70 | 0.02 | 0.74 | 0.02 | 0.77 | 0.01 | 0.68 | 0.01 | 0.53 | 0.02 | 0.56 | 0.05 |
| | | CLIP-KNN | 0.66 | 0.02 | 0.75 | 0.02 | 0.69 | 0.01 | 0.83 | 0.02 | 0.79 | 0.01 | 0.65 | 0.02 | 0.68 | 0.06 |
| | | Dino-Patch | 0.58 | 0.01 | 0.72 | 0.03 | 0.77 | 0.02 | 0.80 | 0.02 | 0.73 | 0.01 | 0.57 | 0.02 | 0.59 | 0.07 |
| | | L-DM | 0.49 | 0.02 | 0.64 | 0.02 | 0.87 | 0.03 | 0.74 | 0.02 | 0.69 | 0.02 | 0.50 | 0.03 | 0.54 | 0.06 |
| | | L-RDM | 0.57 | 0.05 | 0.69 | 0.03 | 0.91 | 0.05 | 0.81 | 0.03 | 0.78 | 0.04 | 0.51 | 0.03 | 0.54 | 0.06 |
| | | Pred-AE | 0.45 | 0.03 | 0.60 | 0.02 | 0.94 | 0.03 | 0.73 | 0.02 | 0.69 | 0.03 | 0.46 | 0.05 | 0.54 | 0.05 |
| | | PredNet | 0.42 | 0.06 | 0.55 | 0.03 | 0.88 | 0.01 | 0.64 | 0.03 | 0.62 | 0.04 | 0.47 | 0.05 | 0.49 | 0.07 |
| | | RES-KNN | 0.52 | 0.02 | 0.65 | 0.03 | 0.78 | 0.02 | 0.74 | 0.02 | 0.69 | 0.02 | 0.53 | 0.02 | 0.55 | 0.06 |
| SAP-Goal2 | strong | AE | 0.84 | 0.03 | 0.86 | 0.04 | 0.52 | 0.04 | 0.92 | 0.03 | 0.91 | 0.02 | 0.54 | 0.02 | 0.71 | 0.08 |
| | | AE-KNN | 0.54 | 0.03 | 0.70 | 0.06 | 0.81 | 0.05 | 0.78 | 0.04 | 0.69 | 0.02 | 0.50 | 0.03 | 0.65 | 0.08 |
| | | CLIP-KNN | 0.62 | 0.02 | 0.76 | 0.05 | 0.75 | 0.03 | 0.83 | 0.03 | 0.76 | 0.02 | 0.56 | 0.02 | 0.72 | 0.07 |
| | | Dino-Patch | 0.49 | 0.03 | 0.70 | 0.06 | 0.84 | 0.04 | 0.77 | 0.04 | 0.65 | 0.02 | 0.51 | 0.03 | 0.69 | 0.07 |
| | | L-DM | 0.61 | 0.05 | 0.73 | 0.06 | 0.75 | 0.05 | 0.81 | 0.04 | 0.75 | 0.03 | 0.52 | 0.02 | 0.68 | 0.08 |
| | | L-RDM | 0.74 | 0.06 | 0.79 | 0.06 | 0.69 | 0.08 | 0.87 | 0.04 | 0.86 | 0.03 | 0.53 | 0.03 | 0.70 | 0.09 |
| | | Pred-AE | 0.37 | 0.16 | 0.64 | 0.07 | 0.95 | 0.02 | 0.73 | 0.05 | 0.59 | 0.12 | 0.39 | 0.20 | 0.67 | 0.11 |
| | | PredNet | 0.46 | 0.03 | 0.64 | 0.07 | 0.89 | 0.02 | 0.70 | 0.05 | 0.62 | 0.02 | 0.54 | 0.05 | 0.67 | 0.07 |
| | | RES-KNN | 0.49 | 0.02 | 0.66 | 0.06 | 0.77 | 0.03 | 0.74 | 0.05 | 0.65 | 0.02 | 0.53 | 0.02 | 0.69 | 0.07 |
| | tiny | AE | 0.80 | 0.10 | 0.79 | 0.06 | 0.53 | 0.08 | 0.87 | 0.06 | 0.89 | 0.08 | 0.52 | 0.03 | 0.49 | 0.04 |
| | | AE-KNN | 0.53 | 0.02 | 0.61 | 0.01 | 0.80 | 0.04 | 0.71 | 0.01 | 0.69 | 0.01 | 0.50 | 0.02 | 0.45 | 0.03 |
| | | CLIP-KNN | 0.62 | 0.01 | 0.68 | 0.01 | 0.73 | 0.02 | 0.78 | 0.01 | 0.77 | 0.00 | 0.55 | 0.01 | 0.53 | 0.04 |
| | | Dino-Patch | 0.49 | 0.05 | 0.62 | 0.03 | 0.81 | 0.05 | 0.71 | 0.03 | 0.66 | 0.04 | 0.51 | 0.03 | 0.47 | 0.04 |
| | | L-DM | 0.55 | 0.06 | 0.63 | 0.03 | 0.79 | 0.04 | 0.74 | 0.03 | 0.73 | 0.05 | 0.49 | 0.02 | 0.45 | 0.03 |
| | | L-RDM | 0.66 | 0.07 | 0.70 | 0.04 | 0.77 | 0.03 | 0.81 | 0.05 | 0.82 | 0.06 | 0.51 | 0.03 | 0.48 | 0.04 |
| | | Pred-AE | 0.46 | 0.04 | 0.56 | 0.02 | 0.92 | 0.05 | 0.68 | 0.01 | 0.68 | 0.02 | 0.50 | 0.02 | 0.48 | 0.03 |
| | | PredNet | 0.44 | 0.05 | 0.52 | 0.02 | 0.85 | 0.10 | 0.61 | 0.02 | 0.61 | 0.04 | 0.49 | 0.02 | 0.43 | 0.03 |
| | | RES-KNN | 0.47 | 0.05 | 0.56 | 0.03 | 0.76 | 0.03 | 0.67 | 0.03 | 0.65 | 0.04 | 0.52 | 0.04 | 0.48 | 0.05 |
| URM-PnP | strong | AE | 0.50 | 0.05 | 0.54 | 0.04 | 0.73 | 0.06 | 0.55 | 0.04 | 0.50 | 0.05 | 0.50 | 0.04 | 0.54 | 0.04 |
| | | AE-KNN | 0.45 | 0.06 | 0.53 | 0.05 | 0.78 | 0.05 | 0.53 | 0.05 | 0.47 | 0.06 | 0.47 | 0.05 | 0.53 | 0.05 |
| | | CLIP-KNN | 0.44 | 0.05 | 0.51 | 0.04 | 0.84 | 0.04 | 0.51 | 0.04 | 0.46 | 0.05 | 0.48 | 0.04 | 0.51 | 0.03 |
| | | Dino-Patch | 0.43 | 0.04 | 0.51 | 0.03 | 0.84 | 0.04 | 0.51 | 0.03 | 0.46 | 0.04 | 0.47 | 0.04 | 0.51 | 0.03 |
| | | L-DM | 0.52 | 0.05 | 0.56 | 0.05 | 0.72 | 0.07 | 0.57 | 0.05 | 0.53 | 0.05 | 0.50 | 0.04 | 0.55 | 0.04 |
| | | L-RDM | 0.62 | 0.05 | 0.59 | 0.03 | 0.51 | 0.07 | 0.60 | 0.03 | 0.63 | 0.04 | 0.55 | 0.03 | 0.55 | 0.04 |
| | | Pred-AE | 0.32 | 0.22 | 0.46 | 0.14 | 0.86 | 0.22 | 0.47 | 0.14 | 0.33 | 0.21 | 0.41 | 0.20 | 0.46 | 0.15 |
| | | PredNet | 0.48 | 0.13 | 0.55 | 0.10 | 0.71 | 0.07 | 0.55 | 0.10 | 0.50 | 0.13 | 0.49 | 0.11 | 0.53 | 0.10 |
| | | RES-KNN | 0.46 | 0.08 | 0.51 | 0.07 | 0.83 | 0.05 | 0.52 | 0.07 | 0.48 | 0.08 | 0.48 | 0.08 | 0.51 | 0.07 |
| | tiny | AE | 0.42 | 0.03 | 0.50 | 0.03 | 0.74 | 0.04 | 0.50 | 0.03 | 0.44 | 0.03 | 0.42 | 0.03 | 0.48 | 0.02 |
| | | AE-KNN | 0.40 | 0.03 | 0.50 | 0.02 | 0.76 | 0.04 | 0.50 | 0.02 | 0.42 | 0.03 | 0.42 | 0.02 | 0.48 | 0.02 |
| | | CLIP-KNN | 0.36 | 0.03 | 0.47 | 0.03 | 0.84 | 0.03 | 0.48 | 0.02 | 0.39 | 0.03 | 0.40 | 0.02 | 0.46 | 0.01 |
| | | Dino-Patch | 0.40 | 0.01 | 0.49 | 0.01 | 0.84 | 0.04 | 0.50 | 0.01 | 0.43 | 0.01 | 0.44 | 0.01 | 0.49 | 0.02 |
| | | L-DM | 0.43 | 0.03 | 0.52 | 0.03 | 0.73 | 0.04 | 0.52 | 0.03 | 0.45 | 0.03 | 0.42 | 0.03 | 0.50 | 0.02 |
| | | L-RDM | 0.61 | 0.02 | 0.59 | 0.01 | 0.49 | 0.04 | 0.59 | 0.01 | 0.62 | 0.02 | 0.53 | 0.01 | 0.53 | 0.02 |
| | | Pred-AE | 0.21 | 0.15 | 0.42 | 0.07 | 0.93 | 0.08 | 0.42 | 0.07 | 0.24 | 0.15 | 0.30 | 0.14 | 0.40 | 0.09 |
| | | PredNet | 0.39 | 0.08 | 0.49 | 0.05 | 0.75 | 0.03 | 0.50 | 0.05 | 0.42 | 0.07 | 0.41 | 0.06 | 0.47 | 0.04 |
| | | RES-KNN | 0.38 | 0.03 | 0.46 | 0.02 | 0.84 | 0.03 | 0.47 | 0.02 | 0.40 | 0.03 | 0.39 | 0.03 | 0.44 | 0.02 |
| URM-Reach | strong | AE | 0.57 | 0.09 | 0.61 | 0.05 | 0.75 | 0.10 | 0.62 | 0.05 | 0.59 | 0.09 | 0.61 | 0.09 | 0.60 | 0.09 |
| | | AE-KNN | 0.52 | 0.05 | 0.58 | 0.03 | 0.81 | 0.05 | 0.59 | 0.03 | 0.54 | 0.05 | 0.55 | 0.05 | 0.56 | 0.05 |
| | | CLIP-KNN | 0.59 | 0.06 | 0.61 | 0.04 | 0.79 | 0.08 | 0.61 | 0.03 | 0.61 | 0.06 | 0.61 | 0.06 | 0.60 | 0.05 |
| | | Dino-Patch | 0.46 | 0.02 | 0.54 | 0.02 | 0.87 | 0.03 | 0.55 | 0.02 | 0.48 | 0.02 | 0.49 | 0.03 | 0.51 | 0.03 |
| | | L-DM | 0.61 | 0.08 | 0.63 | 0.05 | 0.73 | 0.10 | 0.64 | 0.05 | 0.62 | 0.08 | 0.63 | 0.08 | 0.63 | 0.08 |
| | | L-RDM | 0.67 | 0.02 | 0.63 | 0.02 | 0.55 | 0.02 | 0.64 | 0.02 | 0.68 | 0.02 | 0.59 | 0.03 | 0.58 | 0.03 |
| | | Pred-AE | 0.66 | 0.13 | 0.64 | 0.08 | 0.71 | 0.18 | 0.65 | 0.08 | 0.68 | 0.13 | 0.67 | 0.13 | 0.66 | 0.12 |
| | | PredNet | 0.55 | 0.07 | 0.58 | 0.04 | 0.75 | 0.06 | 0.59 | 0.04 | 0.57 | 0.07 | 0.62 | 0.06 | 0.59 | 0.06 |
| | | RES-KNN | 0.61 | 0.06 | 0.60 | 0.04 | 0.76 | 0.09 | 0.61 | 0.04 | 0.63 | 0.06 | 0.61 | 0.07 | 0.60 | 0.07 |

Table 11: Detailed results for all detectors and environments on img observations. The mean and standard deviation are reported for each detector, environment, and anomaly strength. The results are grouped by environment and detector.

| env | strength | detector | AUROC mean | std | AUPR mean | std | FPR95 mean | std | VUSPR mean | std | VUSROC mean | std | Global-AUROC mean | std | Global-AUPR mean | std |
|---|---|---|---|---|---|---|---|---|---|---|---|---|---|---|---|---|
| | tiny | AE | 0.49 | 0.05 | 0.57 | 0.03 | 0.82 | 0.05 | 0.58 | 0.02 | 0.51 | 0.05 | 0.52 | 0.05 | 0.53 | 0.04 |
| | | AE-KNN | 0.48 | 0.01 | 0.56 | 0.01 | 0.83 | 0.03 | 0.57 | 0.01 | 0.50 | 0.01 | 0.50 | 0.01 | 0.52 | 0.01 |
| | | CLIP-KNN | 0.52 | 0.04 | 0.57 | 0.02 | 0.84 | 0.04 | 0.58 | 0.02 | 0.55 | 0.04 | 0.54 | 0.04 | 0.54 | 0.03 |
| | | Dino-Patch | 0.45 | 0.02 | 0.53 | 0.02 | 0.86 | 0.02 | 0.54 | 0.02 | 0.48 | 0.02 | 0.48 | 0.02 | 0.50 | 0.03 |
| | | L-DM | 0.51 | 0.04 | 0.58 | 0.02 | 0.81 | 0.04 | 0.59 | 0.02 | 0.53 | 0.04 | 0.53 | 0.04 | 0.54 | 0.04 |
| | | L-RDM | 0.66 | 0.02 | 0.63 | 0.02 | 0.56 | 0.01 | 0.63 | 0.02 | 0.67 | 0.01 | 0.58 | 0.01 | 0.57 | 0.02 |
| | | Pred-AE | 0.57 | 0.10 | 0.59 | 0.05 | 0.80 | 0.13 | 0.60 | 0.05 | 0.59 | 0.09 | 0.58 | 0.09 | 0.58 | 0.08 |
| | | PredNet | 0.47 | 0.05 | 0.54 | 0.03 | 0.81 | 0.05 | 0.55 | 0.03 | 0.49 | 0.05 | 0.54 | 0.05 | 0.53 | 0.05 |
| | | RES-KNN | 0.54 | 0.03 | 0.57 | 0.02 | 0.82 | 0.05 | 0.58 | 0.02 | 0.56 | 0.03 | 0.54 | 0.03 | 0.54 | 0.03 |
| URR-Reach | strong | AE | 0.47 | 0.08 | 0.55 | 0.06 | 0.75 | 0.07 | 0.56 | 0.05 | 0.49 | 0.08 | 0.47 | 0.08 | 0.47 | 0.06 |
| | | AE-KNN | 0.44 | 0.05 | 0.51 | 0.03 | 0.81 | 0.03 | 0.52 | 0.03 | 0.46 | 0.05 | 0.47 | 0.05 | 0.46 | 0.03 |
| | | CLIP-KNN | 0.55 | 0.02 | 0.53 | 0.02 | 0.81 | 0.02 | 0.53 | 0.02 | 0.57 | 0.02 | 0.54 | 0.03 | 0.51 | 0.03 |
| | | Dino-Patch | 0.49 | 0.04 | 0.51 | 0.02 | 0.84 | 0.02 | 0.52 | 0.02 | 0.51 | 0.04 | 0.49 | 0.03 | 0.48 | 0.02 |
| | | L-DM | 0.64 | 0.06 | 0.61 | 0.04 | 0.64 | 0.09 | 0.62 | 0.04 | 0.65 | 0.06 | 0.59 | 0.08 | 0.55 | 0.08 |
| | | L-RDM | 0.60 | 0.07 | 0.61 | 0.04 | 0.60 | 0.10 | 0.61 | 0.04 | 0.61 | 0.07 | 0.56 | 0.06 | 0.52 | 0.07 |
| | | Pred-AE | 0.43 | 0.09 | 0.48 | 0.05 | 0.87 | 0.06 | 0.49 | 0.04 | 0.45 | 0.09 | 0.48 | 0.09 | 0.46 | 0.06 |
| | | RES-KNN | 0.54 | 0.05 | 0.54 | 0.04 | 0.84 | 0.04 | 0.55 | 0.04 | 0.56 | 0.05 | 0.51 | 0.03 | 0.50 | 0.03 |
| | tiny | AE | 0.49 | 0.04 | 0.56 | 0.03 | 0.75 | 0.04 | 0.57 | 0.03 | 0.51 | 0.04 | 0.48 | 0.04 | 0.48 | 0.03 |
| | | AE-KNN | 0.47 | 0.02 | 0.52 | 0.02 | 0.79 | 0.01 | 0.53 | 0.02 | 0.49 | 0.02 | 0.47 | 0.03 | 0.47 | 0.02 |
| | | CLIP-KNN | 0.57 | 0.04 | 0.54 | 0.03 | 0.80 | 0.04 | 0.55 | 0.03 | 0.59 | 0.03 | 0.55 | 0.03 | 0.52 | 0.03 |
| | | Dino-Patch | 0.50 | 0.04 | 0.52 | 0.03 | 0.85 | 0.02 | 0.53 | 0.03 | 0.52 | 0.04 | 0.49 | 0.02 | 0.49 | 0.02 |
| | | L-DM | 0.63 | 0.06 | 0.60 | 0.03 | 0.66 | 0.08 | 0.61 | 0.03 | 0.64 | 0.06 | 0.57 | 0.07 | 0.53 | 0.05 |
| | | L-RDM | 0.59 | 0.06 | 0.60 | 0.03 | 0.60 | 0.09 | 0.61 | 0.03 | 0.60 | 0.06 | 0.54 | 0.04 | 0.50 | 0.02 |
| | | Pred-AE | 0.42 | 0.03 | 0.48 | 0.01 | 0.90 | 0.04 | 0.49 | 0.01 | 0.44 | 0.03 | 0.47 | 0.03 | 0.46 | 0.01 |
| | | RES-KNN | 0.55 | 0.04 | 0.55 | 0.03 | 0.84 | 0.04 | 0.56 | 0.03 | 0.57 | 0.03 | 0.50 | 0.02 | 0.49 | 0.01 |

