# OpenReview forum: "Anomaly-Gym: A Benchmark for Anomaly Detection in Embodied Agent Environments"
_ICLR.cc/2026/Conference — Submitted to ICLR 2026_

### Official Review · Reviewer_df4P · 2025-10-14

**Soundness:** 3
**Presentation:** 2
**Contribution:** 3
**Rating:** 6
**Confidence:** 5

**Summary:**

The authors introduce Anomaly-Gym, a comprehensive benchmark suite designed to evaluate anomaly detection methods in reinforcement learning (RL) settings. In contrast to existing work, Anomaly-Gym is grounded in principled design criteria that explicitly separate evaluation from algorithmic methodology. By imposing clear constraints on the structure of environments and the nature of anomalies, the benchmark offers a diverse and systematic collection of evaluation data encompassing both simulated and real-world tasks. Overall, Anomaly-Gym comprises 10 environments, 25 anomaly types, 4 anomaly strength levels, and multiple sensor modalities. Extensive experiments on pre-generated datasets highlight the significance of these design choices and their impact on performance assessment.

**Strengths:**

It is non-trivial to establish a clear desiderata that enhances the reproducibility of anomaly detection methods in reinforcement learning (RL). Standardizing a benchmark for this domain is inherently challenging, yet this paper makes a strong contribution by laying a solid foundation for doing so within the RL context. Explicitly:

* I was in particular happy with the attempt of standardizing the agent's policy, which is something other fields do not have to consider. I think the Anomaly Strength is an interesting score to consider, since it empirically gives a level of 'difficulty' with respect to the task relevant features still present in the environment. I can see that this may lead to a more useful difficulty score, especially since the code includes offline data.
* I appreciated the usage of Table 1, simply showing that there is a need for standardizing RL anomaly detection.
* The offline data appears to be well organized, I can see how this can be very accessible, since many benchmarks eventually have RL policies that rely on outdated packaging.

**Weaknesses:**

**Review**

I think that this is a good work, unfortunately what is stopping me from a clear accept is that there is potential for limited impact and that the paper seems unpolished in certain areas.

* A critical issue with this work is that the Real-World RL Suite [1] appears to not be mentioned anywhere in the text. This would perhaps be the largest competitor, although it is an older work, the contribution is slightly reduced since part of the contribution is already available in [1].
* Line 94, I'm not sure a discussion over adversarial RL is necessary. It does not seem to be connected to any of the anomaly types, and there is no mention in the anomaly types about a purposeful malicious injection via AutoAttack or PGD etc.
* The appendix has quite a few typos Ex: (1,) move the cart left/righ,  Page 19 has formating issues, Line 1034 (¡10GB),
* Line 150, Types of Anomalies, seems to be out of place, since Sec 5.2 is where the actual anomalies are discussed.
* Can this be rewritten explicitly: ''Given its interdisciplinary scope (RL, time-series, CV, ...) and the combinatorial space of design
choices (environments, anomalies, onset schemes, ...), several limitations are inherent to this work.''

* Is the AUROC score evaluated on batches of entire trajectories? Or is it computed a different way such as via the score of a balanced dataset of anomaly and non-anomaly transitions? There are many ways to do this, and I feel that this needs to be explicit since there is a lot of variety in the random injections affecting future states and so on in the RL setting. I feel as though there needs to be a short discussion on how the AUROC score was computed.

* Is it possible to do a final review of the citations used and standardize to respective conferences whenever possible ex: [2,3,4,5...] (There are more online, but this is what I could find after a bit of reading).

Some thoughts on potential limited impact:
* It appears most, if not all environments are 'solved' via at least one of the detection methods (PEDM seems to have solved most on its own). I am certain that anomaly detection in the RL setting is far from solved, but is it possible to highlight cases where a major future contribution might exist? Besides image observations (and reducing the tails of the delay scores), it appears that there is no more improvement to be made on the custom datasets, which may significantly reduce the need for this benchmark. I also suspect that Figure 3 could clouding some interesting insights since the easier environments may skew the results here. If so, the paper would benefit from not summarizing all environments into a single figure for better analysis. I also suggest elaborating the image observation results further.
*  Going a bit further in the previous point: 'Our findings
also highlight current challenges with anomaly detection on image data and provide directions for future research.', is a statement I feel the paper needs to improve upon. Section 6.3 does not give much insight to construction of an improved anomaly detector, for example: 'KNN achieves near perfect scores, slightly outperforming all other methods,
showing that method choice can be tailored to failure modality.' does not give the reader a hypothesis on why this might be the case or what could a future reader do to improve, i.e. Why do the tested methods succeed or fail? What do they fail to do? etc.

[1] Dulac-Arnold, G., Levine, N., Mankowitz, D. J., Li, J., Paduraru, C., Gowal, S., and Hester, T. An empirical investigation of the challenges of real-world reinforcement learning. 2020.

[2] Aaqib Parvez Mohammed, & Matias Valdenegro-Toro (2021). Benchmark for Out-of-Distribution Detection in Deep Reinforcement Learning. In Deep RL Workshop NeurIPS 2021.

[3] Jonathan Balloch, Zhiyu Lin, Mustafa Hussain, Aarun Srinivas, Robert Wright, Xiangyu Peng, Julia Kim, and Mark Riedl
NovGrid: A Flexible Grid World for Evaluating Agent Response to Novelty
Proceedings of the AAAI Spring Symposium on Designing Artificial Intelligence for Open Worlds (2022).

[4] Mohamad H Danesh, Alan Fern (2021). Out-of-Distribution Dynamics Detection: RL-Relevant Benchmarks and Results. ICML 2021 Workshop on Uncertainty and Robustness in Deep Learning

[5] Geigh Zollicoffer, Kenneth Eaton, Jonathan C Balloch, Julia Kim, Wei Zhou, Robert Wright, & Mark Riedl (2025). Novelty Detection in Reinforcement Learning with World Models. In Forty-second International Conference on Machine Learning.

[6] Goel, S., Wei, Y., Lymperopoulos, P., Churá, K., Scheutz, M., & Sinapov, J. (2024). NovelGym: A Flexible Ecosystem for Hybrid Planning and Learning Agents Designed for Open Worlds. In Proceedings of the 23rd International Conference on Autonomous Agents and Multiagent Systems (pp. 688–696). International Foundation for Autonomous Agents and Multiagent Systems.

**Questions:**

* Correct me if I am wrong, but it appears that grid-worlds are not considered?
* Is it possible to add a bit more conversation on the choice of anomaly selection? I see the appendix has a bit of this, but I think it would be beneficial on why these anomalies fit the desiderata in the main text.

---

> ### Author Response · Authors · 2025-11-22
> **Answer to Reviewer df4P**
>
> ## Connection to Real-World RL Suite
> Thank you for pointing this out. While the Real-World RL Suite addresses critical challenges, Anomaly-Gym serves a fundamentally different purpose:
>
> Real-World RL Suite (Dulac-Arnold et al., 2020):
> - Goal: Train robust policies under realistic constraints (noise, delays, partial observability)
> - Use case: Policy learning and robustness evaluation
> - No anomaly labels or detection protocols
>
> Anomaly-Gym:
> - Goal: Train under normal conditions, detect when anomalies occur during deployment
> - Use case: Safety monitoring of trained policies
> - Provides: Ground-truth anomaly labels, onset timestamps, calibrated severity levels, detection metrics, and infrastructure to collect data and evaluate AD methods
>
> Nonetheless, there is a connection to this work, as it also includes perturbations to the default environment on similar aspects. We add Real-World RL Suite to Section 2 and clarify how Anomaly-Gym provides the complementary infrastructure for anomaly detection that Real-World RL Suite does not address.
>
> ## Typos, citations, etc.
> Thank you for pointing these out. We will address all of them in the updated version.
>
> ## AUROC computation
> We clarify this in the updated version.
>
> ## AD in RL is far from "solved"
> We absolutely agree that AD in RL is far from “solved”. We would like to clarify several key points:
> PEDM and other methods achieve high scores only for observation anomalies with vector observations (Fig. 2e). This represents one of four anomaly strength levels, one of three anomaly types and one of two sensor modalities. The broader picture reveals substantial open challenges:
>
> Image-based detection (Fig. 2d):
> - High variance across all methods
> - Best method achieves AUROC of 0.60 (far from solved)
>
> Action/dynamics anomalies (Fig. 2b):
> - Simple methods fail
> - PEDM: ~0.87 avg. AUROC (not perfect, still high variance)
> - Environment/anomaly-specific performance varies substantially
>
> Tiny anomalies (Fig. 2c):
> - High variance even for best methods
> - Detection difficulty correlates with policy impact (as intended)
>
> Strong Anomalies
> - Still relatively high variance
>
> Threshold selection (Fig. 3):
> - Long-tailed delay distributions across all methods -> early detection very challenging
> - Major open problem for deployment
>
> In summary: Only a narrow subset (observation anomalies, vector modality) approaches can be described as „solved," which itself provides valuable insight. No method achieves high scores over every setting. The benchmark reveals multiple substantial open challenges that justify its need and will help future research (see below).
>
> ## Aggregation in Figure 3.
> We add individual plots for each domain in the appendix.
>
> ## Current challenges with anomaly detection / Why do the tested methods succeed or fail?
> This discussion indeed fell a bit short as we wanted to focus on the benchmark’s capabilities itself, not the methods applied. However, considering the following three observations:
> - observation perturbations can be directly observed and are thus easier to detect (e.g. even with KNN based approaches directly operating on input vectors)
>  - action/dynamics anomalies can only be observed implicitly or after some interaction and require modeling environment dynamics explicitly. Methods directly operating on observation vectors (e.g. KNN) fail in these settings.
> - current methods/baselines for image observation do not encode environment dynamics beyond simple action concatenation (L-DM)
>
> Based on the above observations, we were able to derive a new improved baseline for image observation.
>
> Our approach extends the AE paradigm by incorporating an RNN within the latent space to model temporal dynamics. Specifically, the model predicts future latent representations conditioned on the current image embedding, the executed action, and the previous hidden state. We term this architecture the Latent Recurrent Dynamics Model (LRDM).
> Although considerably simpler than any transformer-based architecture, LRDM achieves 0.66 +- 0.14 average AUROC over all tasks with image observations. This is an improvement of 6% over CLIP-KNN. This empirically validates the importance of explicitly modeling RL-specific aspects (action and environment dynamics) for robust performance - also in visual observation spaces.
>
> An interesting direction for future work would be to explore more sophisticated encoder architectures (e.g. Transformers, Diffusion models), while explicitly modeling environment dynamics.
>
> ## Gridworlds
> Currently no gridworld environments are considered. This is a deliberate choice, as gridworlds usually have only discrete action/observation spaces.
>
> ### Anomaly Selection
> We add more details on this to in the main text
>
> We thank the reviewer again for their highly constructive feedback. We would like to continue the discussion and answer any remaining questions.

---

### Official Review · Reviewer_HLwf · 2025-10-25

**Soundness:** 3
**Presentation:** 3
**Contribution:** 3
**Rating:** 6
**Confidence:** 2

**Summary:**

The paper introduces a new benchmark for anomaly detection in reinforcement learning, focusing on environments involving embodied agents. The main motivation behind this work is the lack of existing benchmarks for this problem. The proposed benchmark comprises 10 environments, 25 anomaly types and 4 severity levels distributed across various sensor modalities. A pre-generated dataset is also provided. The paper presents a number of classic and neural network-based baselines, evaluated using standard metrics, and provides a thoughtful analysis. Interestingly, the evaluations show that image-based anomalies remain challenging.

**Strengths:**

Benchmark: The paper addresses a research gap. Current benchmarks do not address the issue of anomaly detection in RL, particularly with regard to embodied agents. In this context, the contribution of 10 environments, 25 anomaly types and 4 severity levels is significant.

Evaluation: Furthermore, the evaluated method is clearly distinct from the evaluation protocol. This allows a wide range of approaches to be evaluated using the proposed benchmark.

Baselines: The baseline evaluations are interesting. They demonstrate that simple methods are effective in identifying easy anomalies. They also cover a wide range of methods for vector and image observations.

The paper is well written and easy to follow. The motivation is clear and positioned appropriately alongside the related work. The benchmark is clearly described, and the appendix provides additional helpful details on the environments. Overall, the paper is very readable.

**Weaknesses:**

Real world: Including more real-world evaluation would make a major contribution to the paper. It is certainly not easy, but the current approach is limited. There is the URTdE-Reach evaluation, but it is the only one.

In terms of novelty, the paper does not present any new methods or incremental contributions to existing approaches. While the benchmark is interesting and necessary, it does not include a new method to address challenging situations. Including a method that addresses, even partially, the limitations of existing baselines would be a valuable addition to the paper.

Anomaly difficulty: There is no proof or evidence for Eq. (4). This point needs further clarification.

**Questions:**

How does the proposed benchmark compare with other anomaly benchmarks? Would it not make sense to build on existing benchmarks?

Is there any taxonomy for the 25 anomaly styles?

Does the proposed benchmark allow for real-time interaction, or does it only work offline?

More evaluation on the anomaly annotation would be helpful.

---

> ### Author Response · Authors · 2025-11-22
>
> ## New method
> We thank the reviewer for this feedback. Our goal was to provide an unbiased (decoupled from any method) benchmark, purely designed from principled desiderata.
>
> However, based on our experiments, we were able to derive a new, improved baseline for image observations that improves upon all other methods.
> This approach extends the autoencoder paradigm by incorporating an RNN within the latent space to model temporal dynamics. Specifically, the model predicts future latent representations conditioned on the current image embedding, the executed action, and the previous hidden state. We refer to this architecture as the Latent Recurrent Dynamics Model (LRDM).
>
> LRDM achieves 0.66 +- 0.14 average AUROC over all tasks with image observations. This represents a 6% improvement over CLIP-KNN.
> These results empirically validate the importance of explicitly modeling RL-specific aspects (action and environment dynamics) for robust performance - also in visual observation spaces.
>
> ## Including more real-world evaluation
> We acknowledge this limitation and agree that broader real-world validation would indeed be a valuable addition. We mention this in section 7-Outlook. We also highlight why it is difficult to include more real-world tasks. Specifically, collecting anomalous real-world data is inherently challenging because anomalies are rare and often unsafe to induce: in robotics, they risk hardware damage, and in other domains such as autonomous driving, they can be unethical due to potential harm to humans. Furthermore, controlled, realistic, and reproducible anomaly injection is difficult in many physical systems.
>
> ## Anomaly difficulty
> Equation (4) is a min-max normalization that scales policy performance between two reference points:
> - Upper bound (1.0): Trained policy performance in normal conditions
> - Lower bound (0.0): Random policy performance (worst-case baseline)
> This allows us to express anomaly impact as relative performance degradation independent of environment-specific reward scales.
>
> ## Comparison to other anomaly benchmarks
> Thank you for this question. We appreciate the opportunity to clarify our positioning.
> Table 1 provides a detailed comparison to existing work.  For instance, it shows that existing AD-for-RL benchmarks are fundamentally insufficient in several ways. E.g.:
> - Mohammed & Valdenegro (2021): Classic Gym only (CartPole, MountainCar) with discrete actions
> - Goel et al. (2021), Balloch et al. (2022): Gridworlds with logic-based anomalies not transferable to continuous control
> - Martinez et al. (2024): Analysis paper, not a benchmark
>
> These either lack the scale, diversity, and realism needed for systematic evaluation (Desiderata ED1-ED6, AD1-AD4 in Section 4.2) or are simply not publicly available.
>
> However, we extensively build up on existing work:
> - MJC environments are the same as in (Haider et al., 2023) and originally introduced in (Brockman, 2016). We simply added more anomalies and updated the codebase
> - SAP is built from scratch but partially inspired by openai MPE (Lowe et al. 2020, Multi-Agent Actor-Critic for Mixed Cooperative-Competitive Environment)
> - CAR is built from scratch but partially inspired by (Fu et al., 2021, D4RL) and uses Carla
> - URM is built from scratch but uses Mujoco (Todorov et al., 2012)
> - URRtde is built from scratch but uses an RTDE intferace (Lindvig et al., 2025).
>
> Implementing some environments from scratch was necessary because existing ones could not support the breadth of evaluation needed for the defined desiderata. Our work extends rather than duplicates prior efforts.
>
> ## Taxonomy for anomaly types
> We differentiate action, observation, and dynamics anomalies. Dynamics anomalies follow no taxonomy since they are task-specific. For actions and observations, we define Noise, Scaling, Offset, Drift, Quantization, Temporal Noise, and Delay (see section 5.2).
>
>
> ## Real-Time Interaction
> Yes, all environments also allow for real-time interaction and online RL.
>
> ## Anomaly Annotation
> This is described under section 5.4. Anomalous episodes are generated by introducing a perturbation after a randomized number of steps t_a ∈ (t_0, t_H) (random onset). The timesteps [t_0, ..., t_{a−1}] are labeled as normal, whereas [t_a, ..., t_H] are labeled anomalous.
>
> We thank the reviewer again for their highly constructive feedback. We would like to continue the discussion and answer any remaining questions.

---

### Official Review · Reviewer_kBr5 · 2025-10-29

**Soundness:** 2
**Presentation:** 2
**Contribution:** 1
**Rating:** 2
**Confidence:** 5

**Summary:**

Anomaly detection in reinforcement learning is a critical research area, yet it lacks a unified testbed. This paper proposes Anomaly-Gym, a comprehensive evaluation suite designed for anomaly detection in reinforcement learning settings. Anomaly-Gym comprises 10 distinct environments, 25 anomaly types, and 4 strength levels. The authors implement several baselines, report their performance across these environments, and discuss future research directions.

**Strengths:**

- The paper is clearly motivated and well-written.
- The authors have implemented several baselines, evaluated them across these environments, and provided further discussion—while also pointing out future research directions.
- The code and associated dataset have been made open-source.

**Weaknesses:**

- Anomaly-Gym comprises only 10 environments in total, and most tasks—even CAR-LaneKeep—are relatively simple. Its lack of diversity means it is insufficient to serve as a robust testbed for anomaly detection in reinforcement learning. Furthermore, since some simple methods achieve near-perfect scores on certain tasks, these tasks may not effectively support the design of improved future algorithms.

- The literature review in this paper is not exhaustive. Several highly relevant works are missing, as referenced in [1, 2, 3, 4].

- Table 1 lists some related works on AD for RL. However, these methods are not implemented as baselines in the experiments, which is a major weaknesses for a benchmark-focused paper. Additionally, the code for [5] is publicly available on GitHub.

- The paper provides no explanations for the metrics it uses, such as AU-ROC, AU-PR, FPR95, VUS-ROC, and VUS-PR. This omission prevents the paper from being comprehensive.

[1] Müller, Robert, et al. "Towards anomaly detection in reinforcement learning." Proceedings of the 21st International Conference on Autonomous Agents and Multiagent Systems. 2022.

[2] Wang, Chen, et al. "OIL-AD: An anomaly detection framework for sequential decision sequences." arXiv preprint arXiv:2402.04567 (2024).

[3] Kazari, Kiarash, Ezzeldin Shereen, and György Dán. "Decentralized Anomaly Detection in Cooperative Multi-Agent Reinforcement Learning." IJCAI. 2023.

[4] Sedlmeier, Andreas, et al. "Policy entropy for out-of-distribution classification." International Conference on Artificial Neural Networks. Cham: Springer International Publishing, 2020.

[5] Zhang, Hongming, et al. "A Distance-based Anomaly Detection Framework for Deep Reinforcement Learning." Transactions on Machine Learning Research.

**Questions:**

- The paper refers to these environments as "embodied agent environments," but I find little connection between them and embodied AI. Could the authors explain this classification?

- Please refer to the "Weaknesses" section.

---

> ### Author Response · Authors · 2025-11-22
> **# Answer to Reviewer kBr5**
>
> ## Simple Tasks
>
> We thank the reviewer for this feedback. While we appreciate the critical view, some of the points do not align with our empirical findings.
>
> ### On Task Complexity
> The reviewer's assertion that tasks are "relatively simple" contradicts our empirical findings:
>  1. Image-based detection remains largely unsolved (Fig. 2d): Best methods achieve only ~0.65 AUROC with high variance
>  2. Tiny anomalies are challenging even for the best methods (Fig. 2c): High variance across all detectors
>  3. Timing detection is difficult (Fig. 3): Long-tailed distributions show threshold selection remains an open problem
>  4. CAR-LaneKeep specifically shows substantial variance (0.41-0.68 AUROC across methods, Table 8)
>
> ### On "Simple Methods Achieving Perfect Scores"
> This misinterprets our findings. We show that:
> - KNN achieves near-perfect scores ONLY on observation anomalies (Fig. 2e) - trivial sensor disturbances
> - The same simple methods fail on action/dynamics anomalies (Fig. 2b), where dynamics models are required
> - This demonstrates our benchmark successfully differentiates between trivial and non-trivial detection scenarios
> ### Supporting Future Research
> Our results identify multiple open challenges:
> - Image-based anomaly detection (Sec. 6.3)
> - Robust threshold selection (Sec. 6.3, Fig. 3)
> - Tiny anomaly detection (Fig. 2c)
>
> The benchmark provides the necessary infrastructure to systematically tackle these problems, which was previously impossible.
>
> ## On Benchmark Scale and Diversity
> 10 environments are substantial for this domain. This is also acknowledged by other reviewers. While more is always better, our work already includes:
> - 4 diverse domains: Autonomous driving (CARLA), robotic manipulation (UR-*), robotic control (MuJoCo), and navigation (SAP)
> - 25 anomaly types: across observations, actions, and dynamics
> - 4 calibrated strength levels: tiny, medium, strong, extreme
> - Multiple modalities: Both vector and image observations
> - Real-world validation: URRtde environment with actual robot hardware
>
> As shown in Table 1, no prior work comes close to this scope. Most existing work evaluates on 1-3 simple environments with discrete actions and low-dimensional states. None of the existing works include real-world validation or calibrated anomaly-strength levels.
> Ultimately, we acknowledge several limitations in our current implementation (Section 7) but emphasize that Anomaly-Gym provides a substantial advancement over any prior work.
>
> ## Literature Review non-exhaustive
> We thank the reviewer for providing additional related works. A fully exhaustive overview of all related research was not feasible within the page limit. We instead included the most relevant related articles. Here is more background on why we did not include the mentioned works:
> - [1] only conceptual work (no method, no evaluation, no theoretical/empirical evidence)  blue sky ideas track of AAMAS
> - [2] not a peer-reviewed paper
> - [3] Focus on Multi-Agent Systems and abnormal behavior of compromised agents (different from our setting)
> - [4] No Source Code, evaluation on a single, discrete game environment
>
> Nonetheless, we will include [1] and [4] in the updated version.
>
> ## More Baselines
> We included all methods that are openly available and are applicable to our scenarios. [5] Although publicly available, it only works for discrete action spaces. We refrained from using discrete action spaces, as described in Section 4.2 (ED-3).
> In total, we implemented 17 different baselines, ranging from classical methods to video anomaly detection and transformer architectures. We believe this provides researchers with a good starting point for future development and enables us to gain informative insights into the current capabilities and challenges.
>
> ## Explanations of metrics
> We will include this overview in the updated version.
>
>
> ## Embodied Agents
> Our interpretation of embodied agents aligns with iterature (Liu et al.,2024):
>
> Embodied Agents refer to systems where cognition is integrated into physical entities, such as robots, allowing them to perceive, interact with, and learn from their physical environment. This also includes simulations of such systems.
> In Non-embodied AI systems, cognition and physical entities are disentangled. Examples are stock-trading agents, recommender systems, or Chatbots.
>
> See (Liu et al.,2024, A Comprehensive Survey on Embodied AI)
>
> We thank the reviewer again for their feedback. We hope we have clarified their concerns and would like to continue the discussion.

---

### Official Review · Reviewer_mwsZ · 2025-11-02

**Soundness:** 2
**Presentation:** 2
**Contribution:** 2
**Rating:** 2
**Confidence:** 4

**Summary:**

This paper establishes Anomaly-Gym, a comprehensive evaluation suite for anomaly detection in reinforcement learning settings. The authors design specific constraints on the environments and
anomalies considered that covers a broad spectrum of evaluation data and covers both simulated and real-world tasks.  Anomaly-Gym features 10 different environments, 25 anomaly types, 4 strength levels, as well as multiple sensor modalities. The authors highlight current challenges with anomaly detection on image data and provide directions for future research.

**Strengths:**

1. Overall contribution: The authors make an important contribution to anomaly detection by proposing a principled framework for investigating anomalies. Specifically, they introduce Anomaly-Gym, a suite comprising 10 diverse tasks and 25 types of anomalies, designed to rigorously test, evaluate, and compare different aspects of anomaly detection in reinforcement learning.

2. Empirical evaluation: The authors demonstrate the utility of Anomaly-Gym through a series of experiments, evaluating existing detection methods and baselines across different environments, anomaly types, strength levels, and observation modalities. The results highlight the need to differentiate these factors and reveal the limitations of current approaches, providing valuable insights for future research.

**Weaknesses:**

While the effort to develop such an evaluation platform is highly commendable, I have several concerns regarding its motivation, scope, and validation.

1. **Motivation and scope:** Studying anomalies in a unified manner could indeed be valuable. However, I am not fully convinced by the premise that a centralized platform is necessary or even feasible for investigating anomalies across diverse applications and embodiments in a unified manner. Each application domain exhibits distinct characteristics, making it extremely challenging to design, maintain, and continuously update corresponding environments under a single framework. As a result, it is unclear how this effort could benefit the broader community in a sustainable way, rather than being a one-off implementation.

2. **Missing baselines and related benchmarks:** The paper does not include comparisons with existing methods and benchmarks such as SafeBench [1] and AED [2]. In particular, SafeBench emphasizes safety, a critical aspect of anomaly evaluation, and covers a broader spectrum of driving scenarios than the proposed platform. Including these comparisons would significantly strengthen the empirical foundation and contextual relevance of this work.

    [1] Xu et al., SafeBench: A Benchmarking Platform for Safety Evaluation of Autonomous Vehicles, NeurIPS 2022

    [2] Yeh et al., AED: Adaptable Error Detection for Few-Shot Imitation Policy, NeurIPS 2024

3. **Definition of Anomaly Strengths:** The authors introduce **Anomaly Strengths** as a quantitative measure to compare anomalies across different environments. I believe it is a great attempt. However, this abstraction may oversimplify the notion of anomaly severity. For instance, in SafeBench [1], certain anomalies are safety-critical and could lead to catastrophic outcomes (e.g., loss of life). In such cases, it is difficult to justify reducing anomaly severity to a single-dimensional metric, which may fail to capture the true impact and diversity of anomalies across domains.

4. **Lack of informative insights:** In the conclusion of the experimental section, the authors state that anomaly strength and type significantly affect detection performance, highlighting the challenges associated with image-based observations and threshold selection. However, similar findings could already be derived from existing benchmarks such as SafeBench, without the need to establish a unified framework. Therefore, the added value of the proposed unified benchmark remains unclear.

**Questions:**

1. Is a centralized evaluation platform truly necessary or feasible for investigating anomalies across diverse applications, given the unique characteristics and maintenance challenges of each domain?

2. How does the proposed platform compare empirically to existing benchmarks such as SafeBench and AED, which already cover critical aspects of anomaly detection?

3. Does the Anomaly Strengths metric adequately capture the severity and real-world impact of anomalies, particularly in safety-critical scenarios?

4. What is the added value of the unified benchmark, given that the reported findings could be obtained from existing benchmarks without consolidating multiple environments?

---

> ### Author Response · Authors · 2025-11-22
> **Answer to Reviewer mwsZ**
>
> ## Is a centralized evaluation platform truly necessary?
> ### Necessity:
> - The reviewer suggests that a centralized platform for Anomaly Detection in RL is not necessary.
> - The alternative would be domain-specific evaluation of detection algorithms.
> - As we show in Table 1, this is exactly the current state of research. Different AD methods are presented together with task-specific evaluations in a “one-off” manner
> - This leads to domain-specific, incomparable results.
> - It is currently impossible to assess which AD methods perform well under which conditions.
> - Likewise, we cannot distinguish whether newly proposed methods represent genuine improvements (generalizing across tasks and domains) or whether their improvements stem from overfitting to domain-specific characteristics.
> - Precisely because of this, we argue that a centralized benchmark is necessary.
> ### Maintenance:
> Because Anomaly-Gym is decoupled from any domain or detection method, maintenance is in fact significantly simplified. As mentioned above and in Section 2, current research is fragmented, and codebases are often not openly available. Anomaly-Gym, on the other hand, is open-source and provides a simple, centralized interface for any detection algorithm. It is the opposite of one-off domain-specific implementations. New methods can simply use this interface and our provided evaluation pipeline without any additional overhead. This is also highlighted as a strength by other reviewers.
>
>
> ## Missing baselines and related benchmarks
> ### SafeBech
> We want to clarify a few points:
> - SafeBench is a platform for safety evaluation of autonomous-driving agents. **Note**: AD=Autonomous Driving in this paper, not Anomaly Detection!
> - Evaluation focuses on the performance of driving policies (e.g., collision rate, frequency of running red lights, frequency of running stop signs)
> - Anomaly Detection is not mentioned in this paper
>
> While SafeBench indeed covers a broader spectrum of driving scenarios, it is also limited to such. We therefore believe that SafeBench is only remotely related to our work and does not serve as a suitable baseline. It is unclear to us how to numerically compare Anomaly-Gym to SafeBench.
> ### AED
> This paper introduces Adaptable error detection (AED) in the context of few-shot imitation (FSI) policy learning. The primary goal is to identify instances where FSI policies are inconsistent with the intent of available demonstrations. Crucially, AED expects the availability of failed agent rollouts and precise frame-level error labels during training. This is different from our work. In our work, we consider an unsupervised anomaly detection setting, where only normal samples are available for training.
>
> ## Definition of Anomaly Strengths
>
> We fully agree that anomaly severity is a multifaceted concept, particularly in safety-critical domains where qualitative aspects (e.g., risk of harm) are not easily captured by a single scalar measure.
> However, for the purpose of quantitative benchmarking across heterogeneous tasks, a unified metric is indispensable. In our framework, we therefore define anomaly strength in terms of its effect on the agent’s normalized performance, which provides a consistent and interpretable measure independent of task specifics.
>
> While alternatives such as safety-critical event rate can be meaningful in certain domains, we found that they often produce inconsistent difficulty levels across environments, yielding trivial, easily detectable anomalies in some settings and barely perceptible ones in others (e.g., crashes in Carla vs. collisions in SAP). In contrast, a reward-based definition naturally captures the degree to which an anomaly disrupts a policy, providing a continuous and comparable scale of severity.
>
> Our goal, however, is not to replace domain-specific assessments, but to establish a quantitative baseline that enables consistent comparison of detection performance across heterogeneous tasks. Anomaly-Gym also allows detailed analysis of domain-specific metrics.
>
>
> ## Lack of informative insights
>
> To the best of our knowledge, we are the first to provide empirical evidence supporting these findings (influence of anomaly strength/type and policy on detection performance/time). Existing work only partially covers some of these aspects in limited domains. SafeBench does not evaluate anomaly detection in any aspect. We kindly request that the reviewer point us to a source that supports these findings.
> Also, as the reviewer themselves mention as a strength: “The results highlight the need to differentiate these factors and reveal the limitations of current approaches, providing valuable insights for future research”.
>
> We thank the reviewer again for their feedback. We hope we have clarified their concerns and would like to continue the discussion if there are any remaining questions.

---

### Comment · Area_Chair_LdXy · 2025-11-24
**[ICLR 2026] Author-Reviewer Discussion Phase**

Dear Reviewers,

The authors have posted their rebuttal addressing your concerns. Please kindly review their response, as well as the comments from the other reviewers, and discuss any issues you believe remain unresolved. If the author response does not change your evaluation, please at least provide an acknowledgement indicating that you have carefully reconsidered it.

Thank you again for your dedication and effort in reviewing this submission.
Let’s have a constructive discussion!

Best regards,

Your AC

---

### Meta-Review · Area_Chair_zwxv · 2026-01-12

**Summary:**

This paper contributes Anomaly-Gym - a benchmark for anomaly detection algorithms featuring 10 different environments, 25 anomaly types, 4 strength levels, as well as multiple sensor modalities. The primary concerns raised with the paper are (1) feasibility of coverage across diverse complex real-world applications (2) missing baselines (3) simplified notion of anomaly strengths and (4) somewhat weak in offering insights. The first of these is consistent across the reviews and remain unresolved during the review process. The authors acknowledge: "collecting anomalous real-world data is inherently difficult because anomalies are rare and often unsafe to induce.." -- however this is precisely what is needed for advancing anomaly detection research.

**Reviewer Concerns:**

The primary unresolved concern raised by all reviews is lack of real world coverage that makes insights somewhat inconclusive. It appears that most environments are "easy" as they are somewhat "solved" via atleast one detection method.

**Reviewer Scores:**

The two reject (rating=2) ratings are unlikely to have changed since no new information was provided during the rebuttal.

---

### Decision · Program_Chairs · 2026-01-26

Reject